# Uncovering de novo gene birth in yeast using deep transcriptomics

William R. Blevins [1,2,3], Jorge Ruiz-Orera [1,4], Xavier Messeguer[5], Bernat Blasco-Moreno [6], José Luis Villanueva-Cañas [1,7], Lorena Espinar[2,8], Juana Díez [6], Lucas B. Carey [2,9] & M. Mar Albà [1,10 ✉]

De novo gene origination has been recently established as an important mechanism for the formation of new genes. In organisms with a large genome, intergenic and intronic regions provide plenty of raw material for new transcriptional events to occur, but little is know about how de novo transcripts originate in more densely-packed genomes. Here, we identify 213 de novo originated transcripts in *Saccharomyces cerevisiae* using deep transcriptomics and genomic synteny information from multiple yeast species grown in two different conditions. We find that about half of the de novo transcripts are expressed from regions which already harbor other genes in the opposite orientation; these transcripts show similar expression changes in response to stress as their overlapping counterparts, and some appear to translate small proteins. Thus, a large fraction of de novo genes in yeast are likely to co-evolve with already existing genes.

[1] Evolutionary Genomics Group, Research Programme on Biomedical Informatics, Hospital del Mar Research Institute (IMIM) and Universitat Pompeu Fabra (UPF), Barcelona, Spain. [2] Single Cell Behavior Group, Department of Experimental and Health Sciences, Universitat Pompeu Fabra (UPF), Barcelona, Spain. [3] Single Cell Genomics Group, Centro Nacional de Análisis Genómico (CNAG), Barcelona, Spain. [4] Cardiovascular and Metabolic Sciences, Max Delbrück Center for Molecular Medicine in the Helmholtz Association (MDC), Berlin, Germany. [5] Computer Sciences Department, Universitat Politècnica de Catalunya (UPC), Barcelona, Spain. [6] Molecular Virology group, Department of Experimental and Health Sciences, Universitat Pompeu Fabra (UPF), Barcelona, Spain. [7] Molecular Biology CORE, Hospital Clínic, Universitat de Barcelona, Barcelona, Spain. [8] Department of Gene Regulation, Stem Cells and Cancer, Centre for Regulatory Genomics (CRG), Barcelona, Spain. [9] Center for Quantitative Biology and Peking-Tsinghua Joint Center for Life Sciences, Academy for Advanced Interdisciplinary Studies, Peking University, Beijing, China. [10] Catalan Institution for Research and Advanced Studies (ICREA), Barcelona, Spain. ✉email: malba@imim.es

D e novo gene birth, or the formation of new genes from previously non-coding genomic sequences, has emerged as an important mechanism for the generation of evolutionary novelty[1–3]. In contrast to genes formed by gene duplication or gene fusion, de novo genes have sequences which are unique. Consequently, they can represent veritable leaps of evolutionary innovation. The archetypal version of de novo gene birth begins with a non-genic sequence that undergoes a series of changes, which enable it to be transcribed, translated and potentially confer a new function. While it may seem highly improbable that a few tweaks to non-coding DNA could result in a beneficial new gene, recent evidence has amassed which supports the existence of de novo gene birth across a wide range or organisms[4–14].

The mechanisms driving the initial expression of new transcripts are still poorly understood. Comparative genomics studies indicate that, in mammals, new transcripts can emerge via the chance formation of promoters in intergenic and intronic genomic regions[15]. In other cases, new genes may appear by bidirectional transcription from a conserved promoter[16] or from open chromatin regions near enhancers[17,18]. However, some eukaryotic genomes are very compact and thus have limited intergenic sequences. One such organism is baker's yeast, *Saccharomyces cerevisiae*, in which about 70% of the genome is occupied by coding sequences[19]. It is unclear how this affects the formation of new transcripts. One possibility is that many of the new transcripts overlap existing genes on the opposite DNA strand. Alternatively, they may arise predominantly from bidirectional promoters, as has been observed for the already defined classes of yeast stable unannotated transcripts (SUTs) and cryptic unannotated transcripts (CUTs)[20,21]. In order to answer this question, it is first necessary to identify all transcripts which are expressed in *S. cerevisiae*, including those that are missing from the current gene annotations, and then to compare them to transcripts which are expressed in related species.

Previous studies of de novo gene birth in *S. cerevisiae* have mainly focused on open reading frames (ORFs)[22,23] or annotated genes[10,24]. Here we investigate for the first time de novo gene birth in yeast from the perspective of the transcriptome, using deep RNA sequencing data from *S. cerevisiae* and 10 other species grown in the same two conditions: rich medium and oxidative stress. Additionally, we perform ribosome profiling sequencing of *S. cerevisiae* to determine how many of the de novo generated transcripts encode proteins, using the same two conditions. We use highly specific methods based on read three nucleotide periodicity and homogeneity, to identify bona fide translated ORFs. We also investigate the genomic location of the transcripts with respect to other transcripts. We find that de novo transcripts are strongly enriched in transcripts overlapping other genes in antisense configuration; furthermore, an important fraction of them encode uncharacterized proteins, which may potentially interact with the overlapping sense gene.

## Results

### Identification of over 8000 novel transcripts in 11 yeast species.
The identification of species or lineage-specific genes is based on the comparison of the gene repertoire across different species. However, gene annotations are often incomplete and in the case of *S. cerevisiae* based on ORFs rather than transcripts. To obtain information about possible unannotated transcripts, we assembled transcriptomes for 10 species from the *Saccharomycotina* subphylum including the model organism *S. cerevisiae*, as well as the more distant outgroup species *Schizosaccharomyces pombe*, which were all grown in an identical rich medium (henceforth referred to as normal) and an oxidative stress condition induced

by $H_2O_2$ (henceforth referred to as stress) (Fig. 1a, b, Supplementary Table 1). The transcriptomes were based on a very large number of reads (approximately 60 million reads per species) and covered a wide range of evolutionary distances to *S. cerevisiae* to facilitate the identification of genes of different evolutionary origins. We used the program Trinity to perform de novo transcript assembly, followed by Cuffmerge to obtain a single annotation file for each species that included both annotated and novel transcripts (Fig. 1b, Supplementary Fig. 1)[25]. In total, we identified 8156 novel transcripts across the 11 species (Fig. 1c, Supplementary Table 2). On average, novel transcripts represented 11% of the total transcriptome catalog of each species.

### Discovery of 236 non-annotated putative protein-coding transcripts in *S. cerevisiae*.
To investigate if the novel transcripts in our assemblies contained translated open reading frames (ORFs) we performed ribosome profiling (Ribo-Seq) in *S. cerevisiae* in both normal and stress conditions. Ribo-Seq provides a high-resolution snapshot of where ribosomes are bound; this data can be used to distinguish between stochastic ribosomal association to a mRNA molecule vs. the codon-by-codon ribosomal scanning pattern indicating the active translation of an ORF[26]. Our pipeline, which is based on the detection of nucleotide periodicity and uniformity along the ORF, correctly characterized 97.3% of the verified coding sequences in *S. cerevisiae* as being translated in our samples (Supplementary Fig. 2). Additionally, we identified 236 novel transcripts containing ORFs that showed similar signatures of translation (Fig. 1d). Translated transcripts represented about one third of the novel transcripts identified in *S. cerevisiae*. The newly discovered proteins were much shorter on average than the annotated coding sequences (Fig. 1e). In addition, some of the new transcripts appeared to encode multiple proteins (Fig. 1f).

### Identification of a comprehensive set of de novo originated transcripts in *S. cerevisiae*.
We performed a series of steps to identify which transcripts could have originated de novo (Fig. 2a). First, we used nucleotide and translated nucleotide BLAST homology searches to identify putative homologues in the other yeast transcriptomes; if a transcript had a significant BLAST hit (E-value < 0.05) in another species we considered that the two sequences were likely to share a common origin. Additionally, we inspected the presence of homologues in the proteomes of 35 more distant non-Ascomycota species to discard possible false positives caused by multiple gene loss or horizontal gene transfer (Supplementary Table 3). Second, we identified syntenic genomic regions for the *Saccharomyces sensu stricto* group to detect potential orthologous transcripts whose homology was undetectable with BLAST (Fig. 2b). The percentage of the genome covered by syntenic blocks between pairs of species within the *Saccharomyces* genus ranged from 80 to 91% (Supplementary Table 4). The methodology to identify syntenic regions was based on MUMs, or maximal unique matching subsequences, which provides a solid framework for the effective alignment of genomes[27,28]. If a transcript overlapped another transcript in the same genomic position and strand in another species, we treated the transcripts as potential homologues. Finally, we performed intra-species BLAST homology searches to identify putative paralogues. We estimated ages for each transcript by using the most distant homologous hit, as an estimate of when each transcript had first appeared (Fig. 2c). For example, if the most distant homologous hit for a given *S. cerevisiae* transcript (or any of its paralogs) was in *S. mikatae*, then we estimated that the transcript had emerged sometime after the divergence of *S. kudriavzevii* and

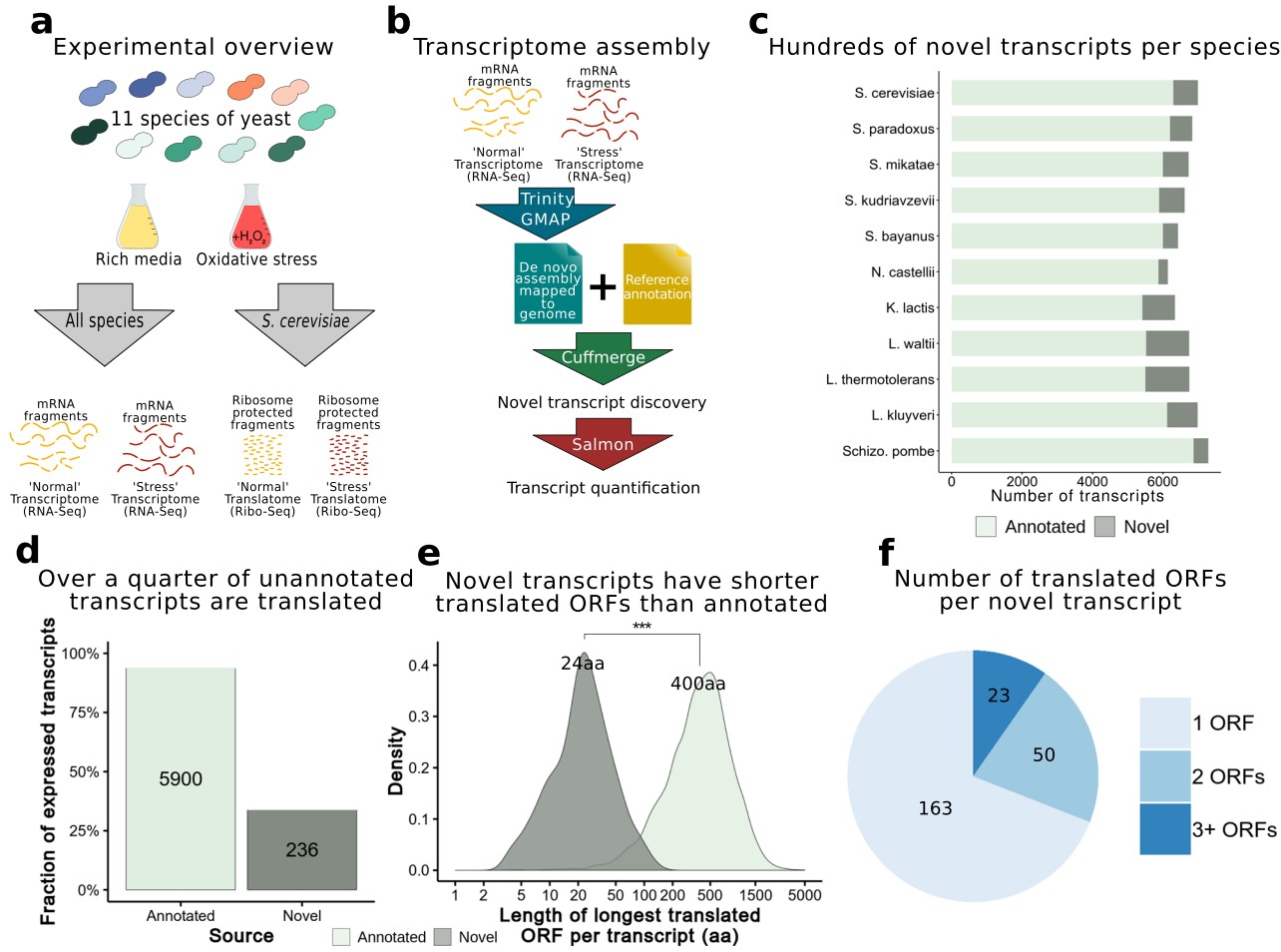

**Fig. 1 Identification of novel, non-annotated, transcripts and proteins. a** Experimental overview of our study. We grew 11 species of yeast in two conditions (rich media and oxidative stress), then performed RNA-Seq on all 22 samples. We also performed Ribo-Seq for *S. cerevisiae*. **b** Transcriptome assembly. We generated a combined transcriptome assembly combining annotated genes together with the subset de novo assembled transcripts not present in the annotations. Subsequently, we quantified the expression of all transcripts in the two conditions. **c** Transcriptomes per species. We obtained hundreds of novel, non-annotated, transcripts, for each species. **d** Prediction of novel translated ORFs. Using the presence of translation signatures in the ribosome profiling data we predicted novel translated ORFs in *S. cerevisiae*. We found 236 non-annotated transcripts likely to encode novel, not yet characterized, proteins. **e** Size of novel and annotated proteins. Novel proteins identified by ribosome profiling were significantly smaller than annotated proteins (two-sided Wilcoxon test, *p*-value < 2.2e-16). Computation of protein length was based on the longest coding sequence per transcript; values in black are the medians. **f** Number of ORFs per transcript. A sizable proportion of the novel transcripts were predicted to encode for more than one ORF. Source data are provided as a Source Data file.

before the divergence of *S. mikatae*, as the most parsimonious scenario.

After applying this pipeline, we selected the transcripts which were expressed over our threshold (>15 TPM, see Methods) in at least one condition, and classified them into three groups: de novo, genus-specific and conserved (Fig. 2d, Supplementary Table 5). In the group of de novo transcripts we included those that were specific to *S. cerevisiae* and those that only had homologues in the closely related species *S. paradoxus* and/or *S. mikatae*; this group was comprised of 213 transcripts, 124 of which were *S. cerevisiae*-specific. We also identified 251 *Saccharomyces* genus-specific transcripts with homology hits in *S. bayanus* and/or *S. kudriavzevii* but not in any more distant species. The rest of transcripts had homologues detected in one or more species outside the *Saccharomyces* genus (conserved, n = 4409). Although some transcripts in the genus-specific and conserved groups may have also emerged de novo, genomic synteny is difficult to trace in these cases and they would be more difficult to validate. The effect of using synteny and paralogs in gene age prediction compared to using only BLAST was not

negligible (Fig. 2d, Supplementary Table 6). For example, if we had not used these additional criteria we would have identified 192 de novo *S. cerevisiae*-specific genes instead of the ones we finally considered valid, 124.

The majority of putative de novo transcripts that we identified did not correspond to annotated genes; 161 out of 213 were previously unannotated transcripts that we would not have identified if we had not performed de novo transcript assembly from RNA-Seq data. The genus-specific transcripts were divided into approximately equal parts of annotated and novel transcripts, whereas the vast majority of conserved transcripts were already annotated (Fig. 2e). Regardless of the conservation class, the number of transcripts expressed solely in normal conditions above our expression level cut-off was clearly larger than those exclusively expressed in stress conditions. This is likely due to the accumulation of mRNAs encoding ribosomal proteins during the response to severe oxidative stress[29], which hampers the detection of lowly expressed transcripts in these conditions. However, there were some indications that, among transcripts detected only in stress conditions, the youngest classes were over-represented; the

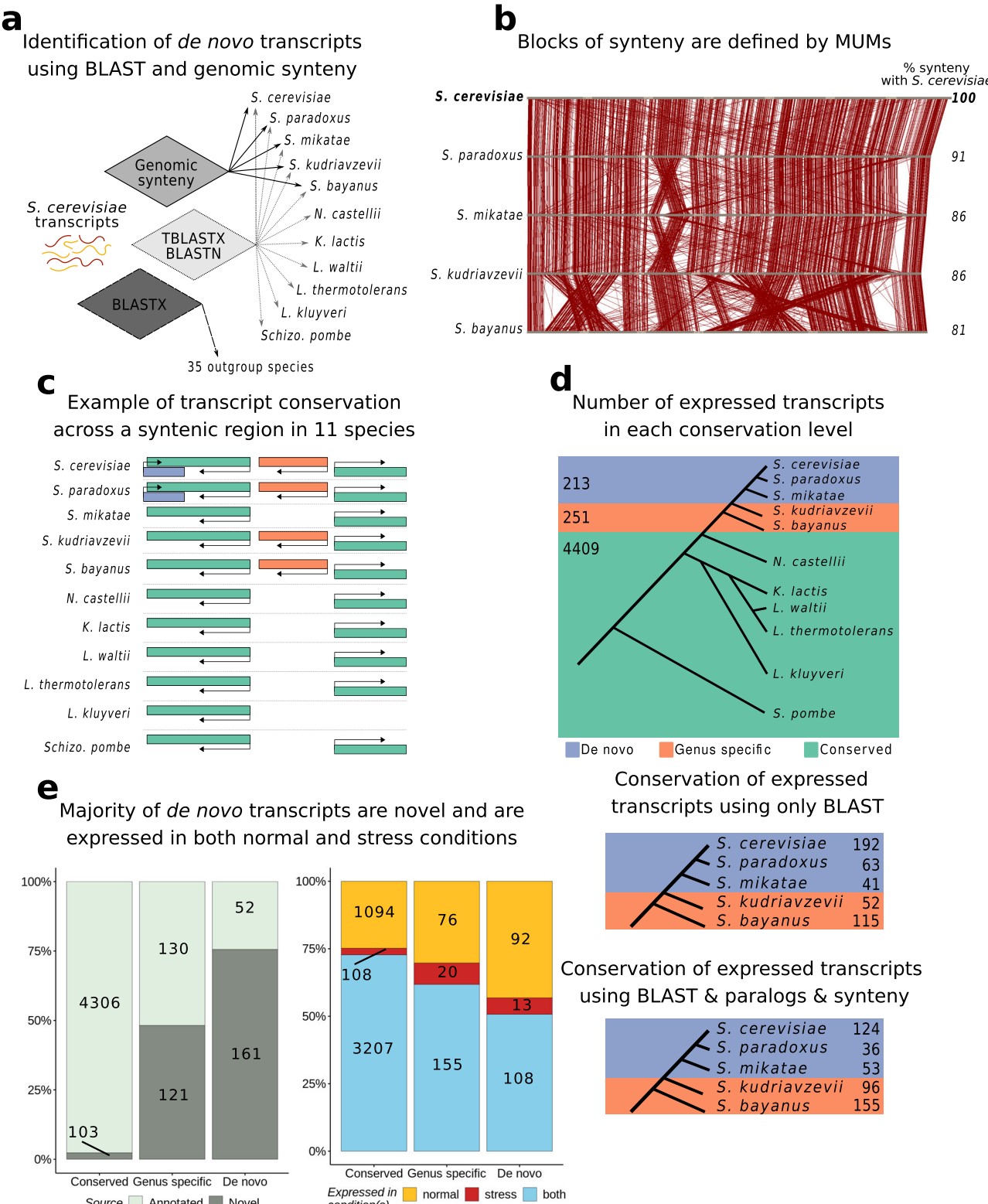

**a** Identification of *de novo* transcripts using BLAST and genomic synteny

**b** Blocks of synteny are defined by MUMs

**c** Example of transcript conservation across a syntenic region in 11 species

**d** Number of expressed transcripts in each conservation level

Conservation of expressed transcripts using only BLAST

Conservation of expressed transcripts using BLAST & paralogs & synteny

**e** Majority of *de novo* transcripts are novel and are expressed in both normal and stress conditions

proportion of de novo and genus-specific transcripts taken together was higher than expected in this subset of transcripts (6–8% observed vs. 2.4% expected, where expected is inferred from the complete set of transcripts, p-value < 0.01 Fisher test).

**Comparison with other approaches**. Our methodology to find de novo transcripts in *S. cerevisiae* was different to the approaches of previous studies; in addition to annotated genes from multiple

species, our study also included thousands of de novo assembled transcripts from 11 yeast species. This allowed us to be very sensitive both at the level of the focal species and at the level of detecting homologues in the other species. To better understand the effect of using transcriptomics data for all species compared, we ran the same computational pipeline but only using transcriptomics data for the focal species (*S. cerevisiae*) or only using annotated genes for all species (Fig. 3a). In the first case we

**Fig. 2 Identification of de novo transcripts in *S. cerevisiae*. a** Pipeline for the identification of de novo transcripts and other conservation classes. For each of the *S. cerevisiae* transcripts which were expressed above our threshold (>15 TPM), we estimated their phylogenetic conservation using genomic synteny and homology searches. **b** Identification of genomic synteny blocks by using MUMs. Diagram illustrating maximal unique matching subsequences (MUMs) across a chromosome in different species of the *Saccharomyces* genus. The synteny blocks were defined by clustering contiguous MUMs in close proximity. **c** Examples of different classes of transcripts depending on their phylogenetic conservation. Diagram of a hypothetical syntenic genomic region shared by all 11 species with different classes of genes indicate. **d** Number of transcripts depending on their phylogenetic conservation. The genes were divided in three classes: 'de novo' (213 transcripts), 'genus-specific' (251 transcripts) and 'conserved' (4,409 transcripts). We found that 213 transcripts were likely to have arisen de novo over the past ~20 million years i.e. there were no homologues in species more distant than *S. mikatae* (purple). Only transcripts expressed at more than 15 TPM were considered here. Below are the number of transcripts identified at each internal branch in the tree leading to *S. cerevisiae*, before and after applying different computational filters. **e** The majority of de novo transcripts are not present in the annotations and are expressed in different conditions. Number of transcripts in each class that correspond to annotated transcripts (light grey) and unannotated transcripts (dark grey). Fraction of transcript expression above 15 TPM in rich media (yellow), in oxidative stress (red), or both conditions (blue). The vast majority of transcripts are either expressed in both conditions or in normal conditions. Source data are provided as a Source Data file.

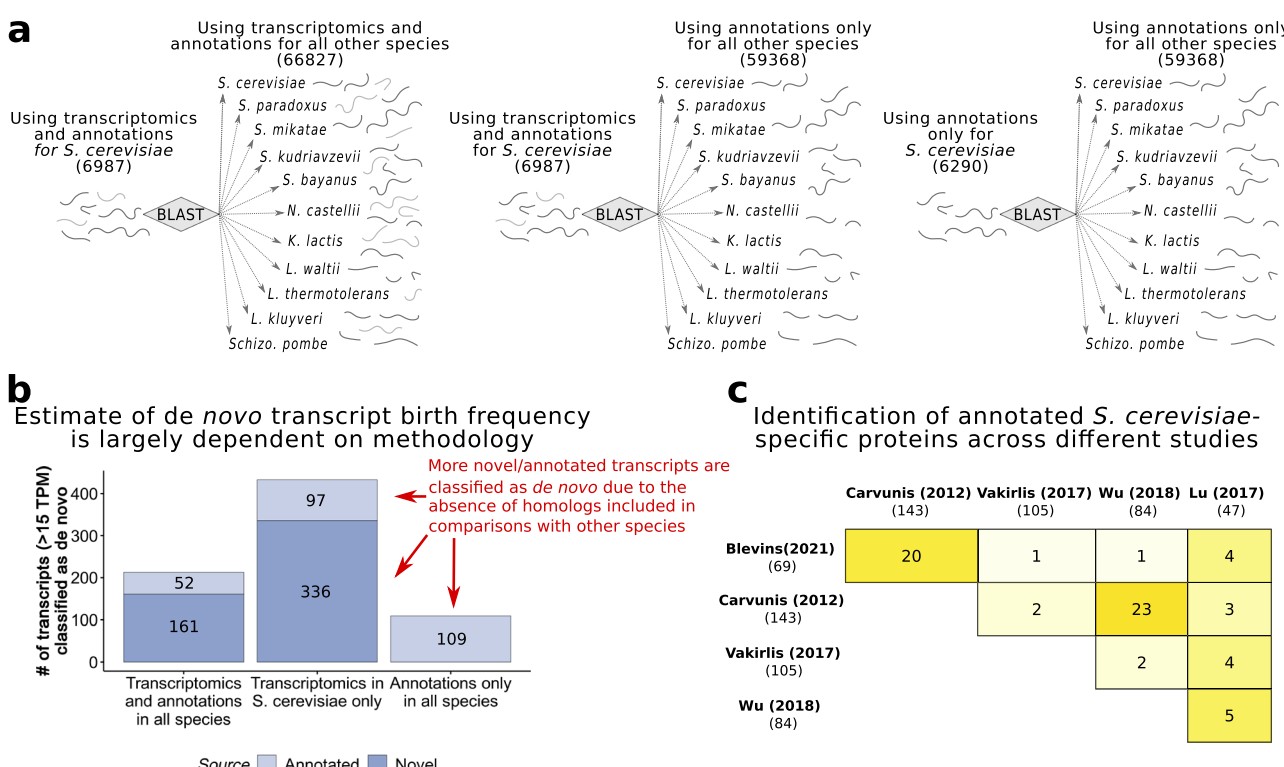

**Fig. 3 Identification of de novo transcripts using different approaches. a** Diagram representing different strategies depending on the use of annotations or transcriptomics data. The use of transcriptomics data to obtain de novo assembled, non-annotated, transcripts, increases the scope of the comparisons of expressed sequences across species. **b** Number of de novo transcripts identified with each of the approaches. The same computational pipeline was applied in all cases, the only differences was whether we used transcriptomics (de novo assembled transcripts) and annotations in all species (our approach), transcriptomics only in the reference species, or annotations only. Many more de novo transcripts were retrieved in the other approaches. **c** Comparison of annotated *S. cerevisiae*-specific de novo proteins that are common between different previously published studies and this study. We used overlap in the genomic coordinates to categorize two transcripts or ORFs as common between two studies. We see moderate overlap in the pairwise comparisons, no two methods have produced very similar results. Note that when several lists existed for the same study we took the least stringent one. Source data are provided as a Source Data file.

obtained a larger number of transcripts classified as de novo (433 vs. 213, Fig. 3b), suggesting that we could overestimate the number of de novo transcripts by two fold in our focal species if we did not include transcriptomics data for the other species. In the second case, we also observed that more genes were classified as de novo than using our approach (109 vs. 52 annotated genes) and of course this approach was missing all the unannotated transcripts which could potentially be classified as de novo.

We also directly compared our results to those of three previous studies, focusing on de novo *S. cerevisiae*-specific annotated protein-coding genes, as this provided a common denominator for the selected studies. The 'Carvunis' study was based on putatively translated ORFs that lacked homologues in other genomes[23], 'Vakirlis' and 'Wu' were based on annotated protein-coding genes[10,24] and 'Lu' on *S. cerevisiae* transcriptomics data[30]. Our analysis was quite conservative compared to some other approaches, and consequently we only classified 69 de novo proteins compared to the range of 47–143 de novo proteins identified in the other studies (Fig. 3c). Despite the differences between the approaches, about one third of the proteins we classified as *S. cerevisiae*-specific were also classified as *S. cerevisiae*-specific in the other studies.

**De novo proteins are small and positively charged.** Considering both the transcripts already annotated as coding and the novel transcripts classified as translated by the analysis of ribosome

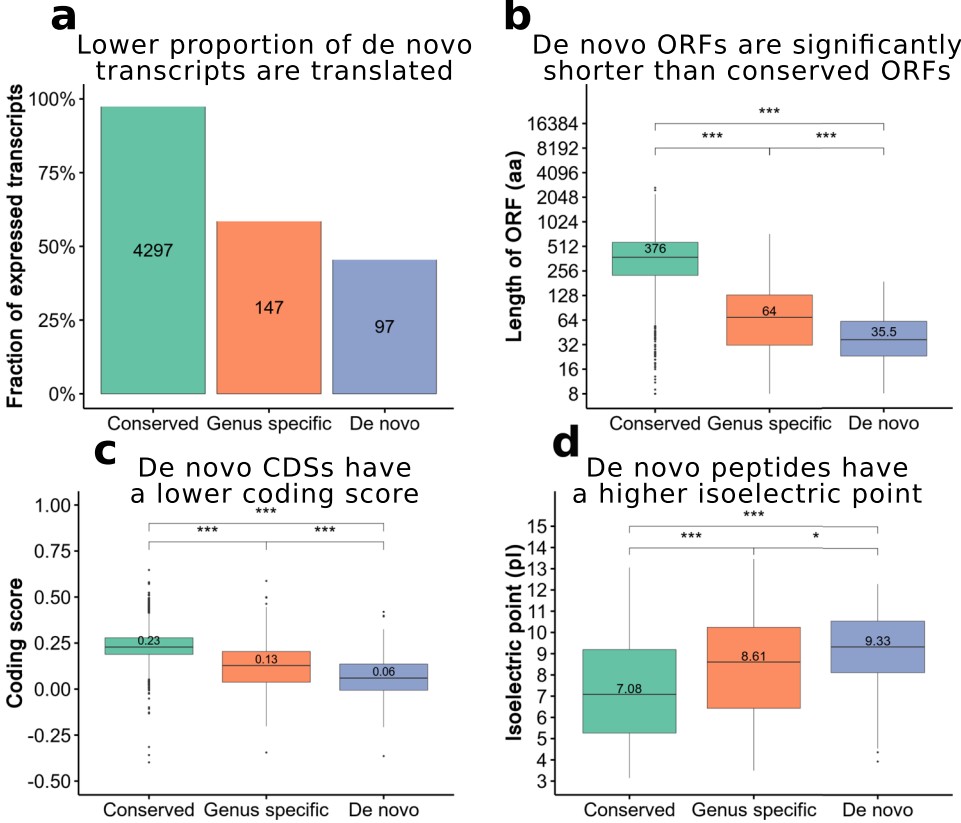

**Fig. 4 Features of de novo proteins. a** Identification of transcripts containing putative translated ORFs using ribosome profiling data and annotations. Using ribosome profiling data from yeast grown in rich media and oxidative stress conditions as well as all the annotated CDS information, we identified 97 de novo, 147 genus-specific and 4297 conserved transcripts with at least one translated open reading frame (ORF). The translated ORFs were detected by RibORF on the basis of high read 3-nucleotide periodicity and uniformity, using a score cut-off of 0.7, in one or both conditions. For sections **b, c,** and **d** we selected the longest translated ORF per transcript (Conserved n = 4297; Genus-specific n = 147; De novo n = 97). **b** Length of translated ORFs in different phylogenetic conservation classes. The length of the longest translated ORF per transcript showed a positive relationship with the conservation level. The median length is indicated in the plot. Length is in amino acids (aa). The values of each boxplot are as follows: 'Conserved' min 5.81, 25% percentile 7.82, median 8.55, 75% percentile 9.16, max 11.13; 'Genus specific' min 3, 25% percentile 4.98, median 6.11, 75% percentile 7.02, max 9.5; 'De novo' min 3, 25% percentile 4.52, median 5.19, 75% percentile 5.94, max 7.55. **c** Coding score of translated ORFs in different phylogenetic conservation classes. Coding score was calculated using a previously developed hexamer-based metric called CIPHER, which measures codon usage bias of putatively coding sequences with respect to non-coding sequences. Coding score shows a significantly positive relationship with the transcript conservation level. The values of each boxplot are as follows: 'Conserved' min 0.05, 25% percentile 0.19, median 0.23, 75% percentile 0.28, max 0.41; 'Genus specific' min −0.2, 25% percentile 0.04, median 0.13, 75% percentile 0.2, max 0.45; 'De novo' min −0.21, 25% percentile −0.01, median 0.06, 75% percentile 0.14, max 0.32. **d** Isoelectric point (IP) of translated ORFs in different phylogenetic conservation classes. IP was predicted with the R package 'Peptides', using the EMBOSS pKscale. Data are for the longest translated ORF per transcript. The values of each boxplot are as follows: 'Conserved' min 3.14, 25% percentile 5.26, median 7.08, 75% percentile 9.19, max 13.06; 'Genus specific' min 3.49, 25% percentile 6.43, median 8.61, 75% percentile 10.24, max 13.46; 'De novo' min 4.54, 25% percentile 8.12, median 9.33, 75% percentile 10.55, max 12.3. Significance between the distributions of the values for different variables was calculated with pairwise two-sided Wilcoxon tests; p-values are as follows: 4b) Gs-C < 2e-16; Dn-C < 2e-16; Dn-Gs <6.3e-05, 4c) Gs-C < 2e-16; Dn-C < 2e-16; Dn-Gs <5.8e-05, 4d) Gs-C 3.9e-06; Dn-C 8.5e-14; Dn-Gs 0.01; where Gs is Genus-specific, Dn is De novo and C is Conserved. Source data are provided as a Source Data file.

profiling data, we identified 45.5% of de novo transcripts (97 out of 213) as having evidence of translation (Fig. 4a). The total number of predicted de novo proteins was 123, as some transcripts contained more than one ORF with signatures of translation. Several de novo proteins also had mass spectrometry evidence; this included four proteins of unknown function ranging in size from 35 to 88 amino acids that had been identified in a large-scale proteomics discovery study[31], as well as the recently described mitochondrial MIN3 protein, which is only 28 amino acids long[32]. The fraction of translated transcripts was over 50% for genus-specific genes, whereas nearly all conserved transcripts were identified as coding (Fig. 4a). One factor to consider is that the lower expression values of the younger genes may have made the identification of potential translation signatures more

difficult. However, when we examined the relationship between expression level and our capacity to detect translation in bona fide proteins, we estimated a sensitivity of the method over 95% in the range of expression of de novo transcripts (Supplementary Fig. 3).

Recently evolved de novo proteins were smaller than more conserved proteins (Fig. 4b). This pattern is expected if these proteins originate from randomly occurring ORFs in the genome and is in agreement with previous observations[1,2,7,23,33–35]. Earlier studies have noted that young coding sequences may not have an optimal codon usage[7,23]. We calculated a coding score, on the basis of a previously developed metric based on hexamer frequencies in coding vs. non-coding sequences[35], in the different groups. In general, the coding scores of the ORFs in the set of de novo transcripts were lower than those of conserved

genes (Fig. 4c), indicating differences in codons usage optimization and/or amino acid composition. Finally, we observed that *S. cerevisiae* de novo proteins had abnormally high isoelectric point values (Fig. 4d). Intriguingly, similar results have been found for mammals[36].

**Genomic location defines different classes of de novo transcripts**. Approximately 70% of the *S. cerevisiae* genome is spanned by annotated coding sequences[19]. The high density of coding sequences in this genome would appear to leave little room for new transcriptional events originating from non-coding genomic sequences. However, there may be an alternative birthplace for new transcripts in baker's yeast; rather than emerging from intronic and intergenic regions, potentially de novo transcripts could arise from the opposite strand of existing coding sequences. To test this, we compared the genomic coordinates of all transcripts to identify those which were overlapping other transcripts on the opposite strand (Fig. 5a).

In accordance with this hypothesis, we found that de novo transcripts were strongly enriched in the subset of transcripts which had antisense overlap, significantly more so relative to conserved genes or genus-specific genes (Fig. 5b, p-value < $10^{-5}$ in both cases using a Fisher test). The majority of these transcripts

were assembled in our pipeline and were not present in the annotations (89 out of 105). Analysis of ERCC spike-in RNA assemblies showed that assembled novel transcripts could not be explained by spurious read orientation or other mapping artifacts (Supplementary Fig. 4). The degree of overlap of de novo transcripts with other genes was very high in most cases and about one third of them overlapped another transcript for the entirety of their length (Supplementary Fig. 5). Considering the cumulative sequences of all 213 de novo transcripts together, 43.4% of the total length of these transcripts overlapped other coding sequences on the opposite strand. We conclude that, in *S. cerevisiae*, many novel transcripts may originate in regions which are protein coding on the other strand.

We also analyzed if there was an excess of de novo transcripts expressed in a divergent orientation from bidirectional promoters, as had been suggested by other studies[10]. We surveyed all pairs of transcripts which were in a divergent orientation and no more than 400nt apart; these transcripts are likely to be separated by a single nucleosome free region[37] (Fig. 5d). We found that 27% of the de novo transcripts were located in a divergent configuration suggesting that they had probably arisen by the activity of an already existing promoter in the opposite orientation. One such example is the already described de novo gene *BSC4*, which may be involved in DNA repair[6]. *ALP1*, an

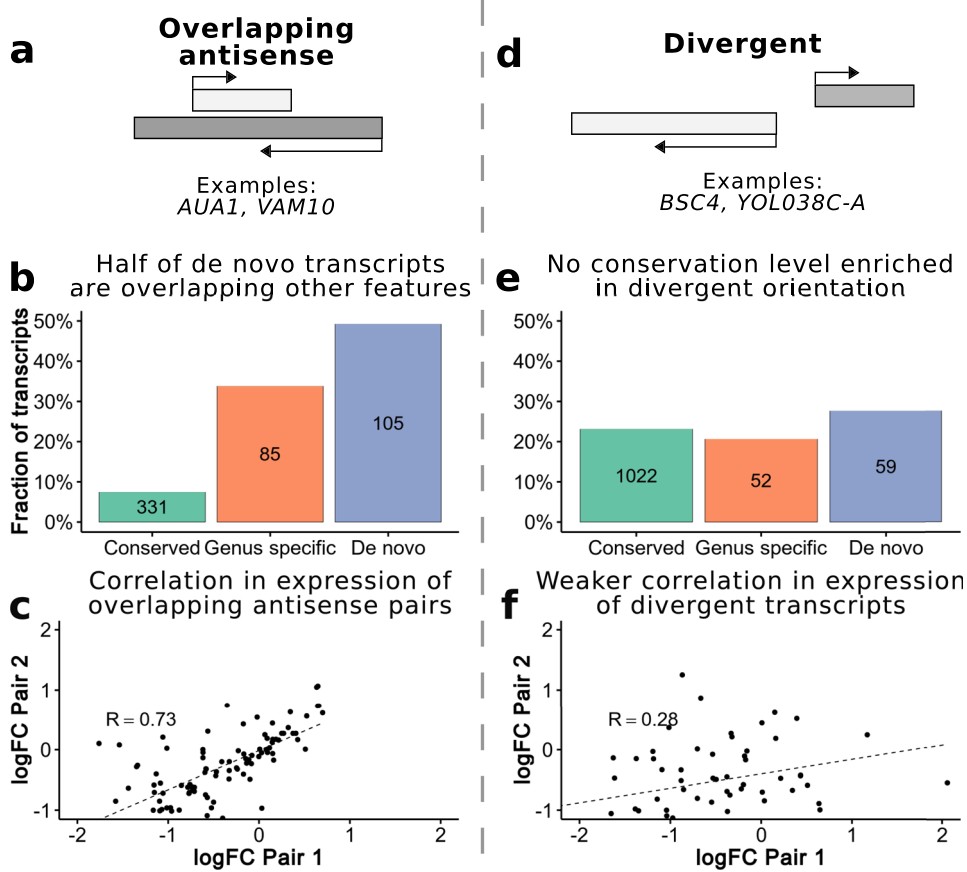

**Fig. 5 Main classes of de novo transcripts in yeast. a** Diagram of a pair of overlapping genes, which are on opposite strands. We consider all pairs of transcripts with any antisense overlap (no minimum overlap threshold). **b** Fraction of transcripts in each conservation level which overlap genes on the opposite strand. A higher proportion of de novo transcripts are in this orientation relative to more conserved transcripts. **c** Correlation fold change (FC) expression values overlapping genes. The differential expression (normal vs. oxidative stress) of gene pairs, which are overlapping each other on opposite strands is strongly correlated (R = 0.73 Spearman's correlation, p-value < $10^{-5}$) suggesting they may be co-regulated. **d** Diagram of a divergent pair of gene. The genes are in a head-to-head orientation on opposite strands and can share a single bidirectional promoter. We considered transcripts separated by 1-400nt to be divergent. **e** Fraction of transcripts in each conservation level, which are in a divergent orientation. **f** Correlation FC expression values divergent genes. The differential expression (normal vs. oxidative stress) of divergent gene pairs is only weakly correlated (R = 0.28 Spearman's correlation, p-value = 0.02754). Source data are provided as a Source Data file.

arginine transporter, is expressed in close proximity in the opposite direction. Another interesting example was *YOL038C-A*, expressed in a divergent orientation to the gene encoding the alpha 4 subunit of the 20S proteasome; we could confirm translation of this 31 predicted amino acid-long protein using the analysis of ribosome profiling data. However, the fraction of transcripts found in a divergent conformation was not higher in the set of de novo transcripts than in other classes (Fig. 5e).

Next, we examined if the changes in the expression of de novo transcripts in stress vs. normal conditions tended to correlate in the two previously described classes of transcripts. In the case of de novo transcripts found in an overlapping orientation we observed a very significant positive correlation in relation to the transcript on the opposite strand (Fig. 5c, R = 0.73, *p*-value < $10^{-5}$). In other words, if the expression of the sense transcript was higher in stress conditions than in normal conditions, the expression of the overlapping antisense transcript showed the same trend, and vice versa. A control in which all transcripts with overlap were randomly paired with a different transcript at a separate locus, that also had overlap with another transcript, indicated that no such correlation was expected by chance (Supplementary Fig. 6). A significant positive correlation was also found in the complete subset of overlapping pairs after excluding de novo transcripts (R = 0.85, *p*-value < $10^{-5}$), indicating that this is not an exclusive feature of de novo overlapping transcripts but that transcripts in this configuration tend to be co-regulated in general. In contrast, changes in expression levels among divergent pairs were only weakly correlated (R = 0.28, *p*-value = 0.027, Fig. 5f, Supplementary Fig. 6).

Finally, we examined whether there were differences in the fraction of transcripts that showed signatures of translation for the two groups, overlapping antisense and divergent. We found no significant differences between the two transcript classes, despite the fact that in the first class the translated ORF often completely overlapped another coding sequence.

### Examples of de novo proteins in sense–antisense pairs.

The only previously well-described example of a de novo gene overlapping another gene in the opposite strand in *S. cerevisiae* is *MDF1*, which overlaps *ADF1*[9,38]. *MDF1* has been proposed to promote vegetative growth and is negatively regulated by the product of *ADF1*. Our transcriptomics-based approach classified *MDF1* as genus-specific (Fig. 6). We also found that, as reported in the original studies, the encoded protein is *S. cerevisiae*-specific, as no comparable ORFs exist in the other species (Supplementary Fig. 7). This provides an example of a de novo gene that may have first been non-coding and only later acquired protein-coding capacity.

Our study identified two other characterized genes that were not previously defined as being de novo and which also overlapped other genes on the opposite strand. The first one was *AUA1*, for which we could no identify any homologues beyond *S. cerevisiae*. The transcript encodes a 94 amino acid-long protein that partially overlaps the gene *WWM1* (Fig. 6). The product of *WWM1* interacts with a caspase-related protease that regulates oxidative stress induced apoptosis[39,40]. *AUA1* mRNA inactivation experiments in yeast indicate that this gene regulates the transport of amino acids across the plasma membrane[41]. Thus, the birth of the gene could have contributed to the adaptation of yeast to high concentrations of ammonia. Interestingly, our RNA-Seq data indicated that the relative expression level of *AUA1* was approximately double in oxidative stress conditions than in normal conditions (118 vs. 59 TPM), suggesting that it may also play a role in oxidative stress response. The second example, *VAM10*, was also found to be *S. cerevisiae*-specific, and according to the literature, it may be involved in maintaining the integrity of

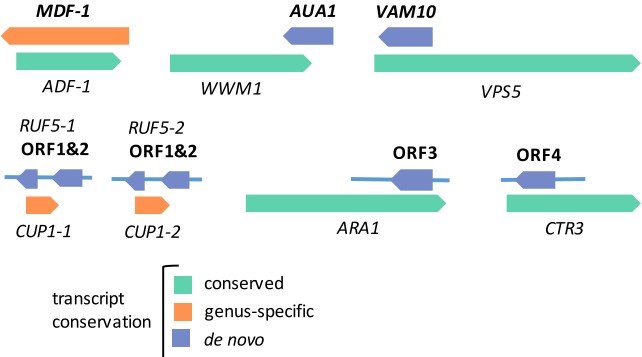

**Fig. 6 Examples of de novo proteins in sense–antisense gene pairs.** The thick arrows represent coding sequences whereas the lines represent the rest of the transcript as defined by our pipeline. ORF denotes open reading frames that are not annotated but which are translated proteins according to the analysis of ribosome profiling data. For convenience the youngest protein (in bold) is always shown in the upper part of the diagram, regardless of whether it is in the positive or negative strand. This representation focuses on the conservation at the level of the transcript. The conservation of the protein may be more restricted as is the case of *MDF1*, which is only found in *S. cerevisiae*.

vacuoles[42]. *VAM10* overlaps *VPS5* on the opposite strand for the entirety of its length; *VPS5* has a role in localizing membrane proteins to the Golgi[43]. In this case, the expression of *VAM10* was above the cut-off of 15 TPM in normal conditions, but not in stress. Interestingly, as in the previous case, the available information suggests that the functions of the two overlapping genes may be related.

We identified other de novo proteins that have not been described in the literature and remain unannotated (Fig. 6, more details of these and other examples can be found in Supplementary Table 7). For instance, we observed two translated ORFs, encoding proteins of size 64 and 37 amino acids, in antisense transcripts overlapping to the two *CUP1* gene copies located in chromosome 8. *CUP1* encodes a metallothionein, which mediates resistance to high concentrations of copper and cadmium[44]. Interestingly, the origin of *CUP1* is also quite recent; our pipeline classified this gene as genus-specific. Both *CUP1* and the newly discovered antisense ORFs were strongly overexpressed in oxidative stress conditions, suggesting that both could be involved in the response to oxidative stress.

Another example was an ORF encoding a 54 amino acid protein overlapping *ARA1* on the opposite strand. *ARA1* encodes a NADP+-dependent arabinose dehydrogenase and a deficient mutant showed increased susceptibility to $H_2O_2$-induced stress[45]. In line with this, we found that the expression of *ARA1* was about double in oxidative stress conditions than in normal growth conditions. A very similar pattern was observed for the new overlapping de novo protein, suggesting that this protein could be involved in the response to stress as well. A third example was a 51 amino acid-long novel protein encoded by a recently originated transcript, which is found overlapping the copper transporter *CTR3*[46]. In this case both proteins were well expressed in rich medium but showed only residual expression (TPM < 15) in oxidative stress conditions.

### ORF conservation vs. transcript conservation.

The identification of homologous transcripts in other species does not imply that the encoded proteins are conserved. Many de novo genes have probably evolved by a transcript-first mechanism, in which the first step is the expression of a non-coding transcript, which subsequently gains an ORF and is translated[47]. The presence of

homologous non-coding transcripts in closely related species has provided evidence of frequent decoupling between the formation of the transcript and the acquisition of coding capacity in *Drosophila*[48] as well as in primates[49].

Our set of de novo transcripts included transcripts that were specific to *S. cerevisiae* as well as transcripts with homologues in *S. paradoxus* and/or *S. mikatae* (but not in more distant species). We took advantage of this classification to examine the conservation of the ORF—for those cases in which the ORF was found to be translated in *S. cerevisiae*—with respect to the conservation of the transcript. We found that for ~84% of the cases in which the transcript was conserved in another species, the ORF was not conserved (Supplementary Fig. 8). This suggests frequent transcript-first formation of de novo genes. We then used the set of *S. cerevisiae*-specific de novo transcripts to estimate the frequency of the ORF-first scenario, in which the ORF would already exist in the corresponding genomic syntenic region of *S. paradoxus* or *S. mikatae* before the transcriptional event. We found that this happened in ~26% of the cases. These results are compatible with the notion that both routes, transcript-first and ORF-first, can contribute to the formation of new genes[47].

## Discussion

Here we compared the transcriptomes of 11 yeast species to identify recently evolved de novo transcripts in *S. cerevisiae*. All species were grown in identical conditions, rich medium and oxidative stress. The use of transcriptomes from multiple species was previously used to investigate de novo gene evolution in *Drosophila*[2], primates[15] and rice[13], but not in the model unicellular eukaryote, *S. cerevisiae*. We wanted to investigate how the compactness of the yeast genome, with 70% of the sequence covered by coding sequences, would impact the formation of new transcripts. We found that *S. cerevisiae* de novo transcripts were strongly enriched in transcripts that overlapped other exons on the opposite strand relative to conserved transcripts with antisense exonic overlap (50% vs. 7.5%, respectively). For comparison, in humans the percentage of de novo transcripts overlapping other exons in the opposite orientation is closer to 10%[15].

Transcripts expressed from bidirectional promoters represented about 27% of the de novo transcripts (23% for conserved transcripts), indicating that bidirectional promoters may not be the main mechanism for the formation of new transcripts, which is in contradiction with the results of a previous study based on annotated genes[10]. Yeast stable and cryptic unannotated transcripts (SUTs and CUTs, respectively) were also reported to be predominantly associated with bidirectional promoters[20,21]. The exception was Xrn1-sensitive unstable transcripts (XUTs); 66% of them were antisense to open reading frames[50].

A sizable fraction of the de novo transcripts we identified in this study are likely to encode proteins, as determined by ribosome profiling experiments performed in the same two conditions as the RNA-Seq experiments. Importantly, we based our predictions on the detection of significant three nucleotide periodicity of the ribosome profiling reads, as opposed to previous approaches based on the number of total reads mapping to the ORFs[23]. This led to the discovery of dozens of non-previously described de novo proteins, many of which are translated from transcripts that are located antisense to other genes.

We used stringent criteria to identify de novo transcripts; any transcripts mapping to the same syntenic region between two species were considered putative homologues. This approach is conservative because observing the expression of two transcripts in the same genomic syntenic region does not necessarily imply

that the transcripts have a common origin, but it is difficult to tell otherwise[51]. Using genomic synteny, as well as extended BLAST searches to include the hits of any paralogs, reduced our initial set of putative de novo transcripts by about one third. The pipeline resulted in the identification of 213 putative de novo transcripts which likely originated over the last 20 million years. This corresponded to about 4.4% of all well-expressed *S. cerevisiae* genes expressed above our threshold (213 out of 4873); if we did not apply our conservative expression cut-off of 15 TPM, this fraction is even higher at 6.2% (436 out of 6986; Supplementary Table 6). We induced oxidative stress to have a sample of the transcriptome in stress conditions for all species; this resulted in the detection of 13 additional de novo transcripts in *S. cerevisiae*, which would not have been found if only using rich medium. Thus, by considering additional experimental settings, the number of de novo transcripts is expected to rise substantially. For example, *BSC4*, a previously identified de novo gene in *S. cerevisiae*[6], was correctly classified into our set of putative de novo transcripts but it was expressed below our expression cut-off of 15 TPM in both of the experimental conditions that we tested.

Despite our conservative criteria to identify recently evolved de novo transcripts in *S. cerevisiae*, the possibility exists that a fraction of them have a more distant origin than the one we inferred. Rapid sequence divergence can make the detection of homologues difficult and result in an underestimation of the age of some genes[52–54]. However, sequence evolution simulations indicate that this should have only a minor effect in comparisons of closely related species such as those within the *Saccharomyces* genus[55]. In addition, synteny-based analytical studies have recently shown that most genes for which we fail to detect homologues in more distant species, or orphans, are likely to have arisen de novo[51]. Perhaps more importantly, here we used genomic synteny in addition to sequence similarity searches across the transcripts, which should reduce even further the number of possible false positives.

Another possible source of errors in the identification of the branch of origin of a gene is the loss of a given transcript in one or more species. Let's imagine that a transcript originated in the common branch of the *Saccharomyces* genus but was subsequently lost in *S. bayanus* and *S. kudriazevii*, being currently only present in *S. cerevisiae*, *S. paradoxus* and *S. mikatae*; this transcript would have been classified as de novo by our pipeline, when it is actually genus-specific. As genes of different ages may be lost at different frequencies[56], it is difficult to estimate how often such loses may have occurred. We dealt with this uncertainty by creating classes that were larger than a single internal branch and which grouped several branches and species; these classes were more robust to errors caused by secondary loses of genes, especially if this happened in a single species. Finally, we also have to consider that a transcript may completely change its expression pattern in one or more species, and become undetectable when using the same conditions for all species. This phenomenon is probably relatively rare and the transcript would likely still have some basal expression levels in rich medium. To this point, we observed that 95% of the *S. cerevisiae* annotated genes could be detected as expressed in rich medium above our lower limit of detection (TPM > 2). We also have to consider that we used high sequencing coverage, which facilitates the detection of lowly expressed genes.

Previous studies to investigate de novo gene evolution in *S. cerevisiae* focused on ORFs rather than transcripts[10,23] or on transcriptomics data for *S. cerevisiae* only[30]. We investigated the impact that using only gene annotations, or only using transcriptomics data for the focal species, would have in the results obtained with our pipeline. We observed that these two approaches could result in an increase in the number of false positives because of missing data in other species.

An appealing hypothesis regarding the role of de novo genes is that their emergence may be associated with increased adaptation to environmental stresses[57]. Several prior studies in yeast have found that a significant fraction of putative de novo genes are expressed in starvation conditions when compared to rich media[22–24]. Here we found that transcripts exclusively expressed in the stress condition were enriched among the youngest classes of genes—de novo and genus-specific—supporting this idea. While this enrichment was modest, the question merits further investigation as it could provide clues into the functions of a subset of de novo transcripts.

Although our study was centered on de novo transcripts rather than ORFs, we also investigated how many of these transcripts could translate small proteins using ribosome profiling data generated for the same conditions. It is worth noting that many small proteins are likely missing from the reference set of annotated coding genes because they can neither be differentiated from randomly occurring ORFs by computational means[58] nor can they be detected by traditional proteomic approaches[59]. The development of ribosome profiling techniques has provided a way to overcome these limitations, especially when combined with three nucleotide periodicity patterns in the sequencing reads, which ensure the identification of bona fide translation events[26,60]. Using ribosome profiling, our study identified 97 de novo transcripts that contained ORFs with clear evidence of translation. A recent study in the free-living related species S. paradoxus has also uncovered many novel translatable ORFs using ribosome profiling data[61]. These studies illustrate the existence of a very large unexplored set of proteins that may underlie many of the recent adaptations in yeast species.

Surprisingly, the proportion of de novo transcripts that contain ORFs with evidence of translation was similar for overlapping antisense and for intergenic transcripts. This is remarkable because, in the first case, the newly evolved ORFs were often completely embedded in the coding sequence of the other strand and changes in one coding sequence may be deleterious for the other coding sequence. It has been proposed that the ancestral class I and II aminoacyl-tRNA synthetases evolved from complementary strands of the same locus[62,63] and one already characterized de novo gene in yeast, MDF1, is also overlapping another gene, ADF1[9]. Our study provides abundant new material for investigating the co-evolution of such overlapping coding sequences.

In addition to new coding transcripts, we identified a large number of de novo transcripts which appear to be non-coding, as they did not display signatures of translation. As the previous studies on de novo genes in yeast have focused on ORFs, this type of de novo transcripts have remained understudied. Some antisense transcripts may play a role in controlling the abundance of the protein encoded by the sense gene[64,65]. Huber et al. 2016 repressed antisense transcripts of 162 yeast genes and observed an effect in about 25% of the genes, mostly a weak decrease in the amount of the sense protein[66]. Here we observed that changes in the expression of sense and antisense genes tended to be positively correlated. On the basis of this, and on specific observations for gene pairs with experimental information for both members of the pair (AUA1-WWM1, VAM10-VPS5 and MDF1-ADF1), we can speculate that the functions of de novo proteins in antisense transcripts may often be related to that of the overlapped gene, by being involved in related cellular processes or by regulating the activity of the gene. An interesting example was a de novo ORF encoding a protein of 64 amino acids that overlapped CUP1, a metallothionein-encoding gene. The expression of the two transcripts of the sense–antisense pair increased about two fold under oxidative stress conditions (from 246–304 to 570–700 TPM), suggesting that both proteins may have a role in the response to stress.

This work establishes, using transcriptomics data from multiple species and genomic synteny, that about 5% of the baker's yeast transcriptome has arisen de novo fairly recently. We have found that a disproportionately large fraction of these transcripts are overlapping other genes on the opposite strand, showing that this could be a main route for the evolution of de novo genes in species with compact genomes i.e. with relatively small fractions of intergenic or intronic sequences. Additionally, we propose that this genomic configuration can enhance the functionalization of the new transcripts, which could inherit regulatory features from the older overlapped gene. As this configuration is not so common in more conserved transcripts, this antisense overlap may be beneficial for relatively short timescale adaptations (in the order of tens of millions of years). Large-scale experimental transcript inactivation screenings coupled with the monitoring of gene expression changes may provide new clues to their possible regulatory activities or their involvement in increased organism survival in the face of environmental challenges.

## Methods

**Yeast material**. The 11 yeast strains used in our analysis (Supplementary Table 1) were selected due to their phylogenetic distribution and their ability to grow in the two conditions tested (see below). Several species, which are closely related to S. cerevisiae, were included to facilitate genomic synteny comparisons. A group of more distant and sparsely distributed species was included as well to broaden the scope of the homology searches. Yeast strains were obtained from the labs of both Lucas Carey and Kevin Verstrepen. We used S288C strain of S. cerevisiae, Genbank genome entry GCF_000146045.2. More information about all strains used in our experiments is available in Supplementary Table 1.

**Experimental conditions**. We opted for growth conditions that would accommodate many species of yeast[67]; all 11 strains were grown in a custom rich media at 30 °C (Supplementary Fig. 9). For each species, we selected an isogenic population from streaked plates, then incubated cultures overnight. We used the overnight culture to inoculate two identical 125 mL Erlenmeyer flasks containing 20 mL of rich media each. After approximately 6 generations of log phase growth, around $OD_{600}$ of 0.3, we added $H_2O_2$ to one flask to a final concentration of 1.5 mM; after 30 min, the yeast cells were harvested and frozen from both the stressed and the unstressed flask. We chose a concentration of 1.5 mM hydrogen peroxide as we found that this concentration would approximately halve the growth rate for the species included in our study (Supplementary Fig. 10); a treatment period of 30 min of $H_2O_2$was selected to capture the greatest variation in expression during stress response[68]. For each sample, four 1.5 mL centrifuge tubes of cell culture were extracted, centrifuged at 4 °C, and then frozen at −80 °C. This protocol was slightly modified for the ribosome profiling experiments to account for the increased demand in raw material for the sequencing protocol (see Ribosome profiling section below).

**Transcriptomes**. We performed strand-specific RNA sequencing of 11 species of yeast grown in rich medium and oxidative conditions on a Illumina sequencing platform. The total number of mapped reads was between 28 and 38 million reads per sample. The transcriptomes were assembled using a pipeline that included Trinity for de novo transcript assembly[69], Transrate to evaluate the quality of each assembly and refined the parameters of Trinity to achieve a high-quality de novo assembly[70], GMAP to map the assembled transcripts back to the reference genome[71] and, Cuffmerge from the Cufflinks suite version 2.2.0 to combine the de novo assemblies from normal and stress conditions with the reference transcriptome[72] (Supplementary Fig. 1). When we combined novel and annotated transcripts into a comprehensive transcriptome, novel transcripts from our assembly which overlapped the reference annotations were considered redundant and eliminated; however, these transcripts were still included in the BLAST database during homology searches. More details on the transcript assembly pipeline have been published elsewhere[25].

**Determination of a gene expression cut-off for comparative transcriptomics**. In order to compare the transcriptomes of different species we first needed to establish which was the transcript gene expression threshold that would guarantee that the transcripts could be assembled from the RNA-Seq data. During the library preparation step we had added synthetic spike-in transcripts from the ERCC spike-in kit to each sample. This spike-in allowed us to determine that complete and reliable de novo assembly of a transcript could be achieved when the expression of the transcript was above 15 transcripts per million units or TPM (Supplementary Fig. 11). We also established the lower limit of detection of a transcript already present in the annotations, TPM > 2. We identified 4873 transcripts in S. cerevisiae which were expressed above the 15 TPM cut-off in at least one of the two

conditions tested, including 4488 annotated and 385 novel transcripts (Supplementary Table 5, Supplementary Fig. 12). For the other species we did not use any expression cut-off to be as sensitive as possible in the sequence similarity searches.

**Ribosome profiling**. Cultures were grown in 500 ml of rich media in 1 L Erlenmeyer flasks; we added cyclohexamide (100 μg/ml final concentration) 1 min prior to harvesting the cells. We harvested the yeast cells via vacuum filtration, suspended them in 500 μl of lysis buffer, then flash-froze them with $N_2(l)$. For each sample, 2/3 of the harvested cells were reserved for Ribo-Seq and 1/3 for RNA-Seq. Cells were lysed using the freezer/mill method (SPEX SamplePrep); after preliminary preparations, lysates were treated with RNaseI (Ambion), and subsequently with SUPERaseIn (Ambion). Digested extracts were loaded in 7–47% sucrose gradients to evaluate the quality of the samples. Monosomal fractions corresponding to digested polysomes were collected; SDS was added to stop any possible RNAse activity, then samples were flash-frozen with $N_2(l)$. RNA was isolated from monosomal fractions using the hot acid phenol method. Ribosome-Protected Fragments (RPFs) were selected by isolating RNA fragments of 28-32 nucleotides (nt) using gel electrophoresis. The protocol described in Ingolia et al. 2012 was used to prepare sequencing libraries for both RPFs and fragmented RNA, with minor modifications[73]. Sequencing was performed on the Illumina NextSeq platform. We performed strand-specific sequencing, which permits the differentiation between the products of sense and antisense overlapping sequences.

**BLAST homology searches**. The transcripts from each species were subjected to an all-against-all homology search using BLASTN and TBLASTX, not considering matches on the opposite strand, and an e-value cut-off of 0.05[74]. BLASTX homology search was also performed against the proteomes of 35 distant species. The BLAST databases contained all annotated as well as novel transcripts from our assemblies, without any expression cut-off. With regards to BLASTN, we only considered hits whose alignment was over 100nt. Homologous transcripts found in the same species were treated as paralogs; we recorded the most distant homology hit for all paralogs of a given transcript. This allowed us to infer potentially deeper conservation for all copies of duplicated genes.

**Genomic synteny comparisons**. Syntenic genomic regions in pairs of species were identified with an adapted version of M-GCAT[28]. The program searches for significant seeds of identical sequences between two genomes called MUMs (maximal unique matches), then sets of parallel, consecutive, and neighboring MUMs are clustered into synteny blocks. We used a maximum distance of 100 bases to cluster two consecutive MUMs. We used the information on the genomic coordinates of the MUMs in the pair of species compared to assess if there was overlap between any two transcripts in two different genomes. More specifically, for each transcript in the first genome we first determined whether it was included in a MUM cluster, by comparing the coordinates in the GTF file with those in the clusters, and then used the MUM coordinates located just before and after the gene to recover the corresponding coordinates in the second genome. We could identify regions of conserved synteny in other species from the *Saccharomyces* genus for the vast majority of the transcripts. If available, we used this information to check if there was any transcript expressed in the second genome whose genomic location overlapped the segment between those coordinates. Transcripts overlapping the same syntenic region were treated as potential homologues.

**Prediction of translated ORFs**. We used an in-house script to generate genomic coordinates for all possible ORFs for each transcript; this script scans the transcript for canonical and non-canonical start and stop codons, then returns all ORFs with >3 codons long and not fully contained in a longer ORF in the same frame. We used RibORF[75] to analyze our Ribo-Seq data using the parameters of minimum length = 9aa, minimum number of reads = 10. RibORF counts the number of reads that fall in each frame and calculates the distribution of reads along the length of the ORF. We used a RibORF score cut-off of 0.7, as proposed in the original study, to predict translated ORFs. We considered an ORF as translated if we observed a RibORF score >0.7 in either normal, stress, or both conditions, independent of the condition(s) in which transcription >15 TPM occurred. The same applied for transcripts with multiple ORFs with evidence of translation. The vast majority of annotated coding sequences with 10 or more mapped Ribo-Seq reads were classified as translated using this cut-off (97.3%), indicating high sensitivity of the method (Supplementary Fig. 2). The false positive rate of the method was previously estimated to be 3.33% using the same parameters as those employed here[35].

**ORF properties**. We quantified several properties of translated ORFs; these ORFs comprised the sequences annotated as protein coding as well as the ORFs in novel transcripts predicted to be translated by RibORF (see above). The coding score of coding sequences/ORFs was calculated using a previously developed hexamer-based metric called CIPHER[35]. The method uses a table of pre-calculated hexamer scores that measures the relative frequency of each hexamer in coding vs. non-coding sequences in different species, including *S. cerevisiae*. The coding score is the average value of all possible in-frame hexamers in the sequence. CIPHER is

available at https://github.com/jorruior/CIPHER. The protein isoelectric point (IP) was predicted with the R package 'Peptides', using the EMBOSS pKscale[76].

**ORF conservation analysis**. For each de novo transcript with translation evidence or that was annotated as coding (99 transcripts), we generated a multiple sequence alignment with Clustal Omega[77] that included the corresponding genomic region in *S. cerevisiae* as well as the syntenic regions of *S. paradoxus* and *S. mikatae*. We annotated the relative position of the translated ORF/s in *S. cerevisiae* as well as compiling a list of all possible peptide sequences that could arise from all ORFs (ATG to STOP codon) in the syntenic sequences of *S. paradoxus* and/or *S. mikatae*. To determine if an ORF was conserved we manually inspected the alignments and the peptide sequences. Conserved ORFs were those that corresponded to the same genomic location and were at least half the length of the translated ORF in *S. cerevisiae*. We applied a similar procedure to study the conservation of the ORF encoding the MDF1 protein, but in this case we also included *S. kudriavzevii* and *S. bayanus* in the comparison.

**Reporting summary**. Further information on research design is available in the Nature Research Reporting Summary linked to this article.

## Data availability
The raw RNA sequencing data can be freely and openly accessed on the Sequence Read Archive (SRA) with project ID SRP187756. Transcript assemblies can be downloaded from https://doi.org/10.6084/m9.figshare.7851521.v2. The raw ribosome profiling (Ribo-Seq) data are found under BioProject number PRJNA435567. The data used for the analyses are available at https://doi.org/10.5281/zenodo.4321014. Source data are provided with this paper.

## Code availability
The code to generate the figures is available at https://doi.org/10.5281/zenodo.4321014.

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

## Acknowledgements

We thank Dr. Ksenia Pugach and the Verstrepen lab for cultures of several species of yeast, Leire de Campos-Mata for assistance with the preparation of the RNA for

sequencing, and the Sequencing Facilities at the Center for Regulatory Genomics (CRG) and Universitat Pompeu Fabra (UPF). The work was funded by grants PGC2018-094091-B-I00, BFU2015-65235-P, BFU2015-68351-P, BFU2016-80039-R, TIN2015-69175-C4-3-R and RTI2018-094403-B-C33 from Spanish Government—FEDER (EU), and from grant PT17/0009/0014 from Instituto de Salud Carlos III—FEDER. We also received funding from the "Maria de Maeztu" Programme for Units of Excellence in R&D (MDM-2014-0370) and from Agència de Gestió d'Ajuts Universitaris i de Recerca Generalitat de Catalunya (AGAUR), grants number 2014SGR1121, 2014SGR0974, 2017SGR1054 and 2017SGR01020 and, predoctoral fellowship (FI) to W.R.B.

## Author contributions

W.R.B. obtained the samples, designed the pipeline and performed most data analyses. J.R-O. analyzed ribosome profiling data. X.M. analyzed genomic syntenic regions. B.B-M. performed ribosome profiling experiments. J-L.V-C. assisted in the design of the pipeline. L.E. assisted in sample preparation. J.D. supervised ribosome profiling experiments. L.B.C. and M.M.A. supervised the project. W.R.B. and M.M.A. wrote the paper with input from all authors.

## Competing interests

The authors declare no competing interests.
