## [Peer Review File · Nature Communications]

Reviewers' Comments:

Reviewer #1:

Remarks to the Author:

The authors perform a RNAseq experiment on 11 yeast species spanning billions of years of evolution. They use these RNAseq experiments (1 replicate per species) and ribosomal profiling experiments (in *S. cerevisiae* only) to identify putative de novo genes. They focus their detailed analyses on *S. cerevisiae*. They find that antisense transcripts (on opposite strands from a coding sequence) may often contain de novo genes. There is no functional follow up on any of the findings to confirm expression and translation.

The work has been well performed and will provide a useful resource for the community and a very rich dataset to mine in the future. I do not see noticeable limitations to the methods and the analysis. However, the work is very descriptive and brings limited new findings to the field. The manuscript appears not to be considering all of the previous work that has been done on yeast de novo genes. This would have helped see how their work brings the field forward. They do cite the relevant papers but do not really compare their results with previous papers examining de novo genes and their properties so we do not see what are the novel observations that they made. In the abstract, the authors mention "However, little is known about the formation of new transcripts in more densely-packed genomes such as *Saccharomyces cerevisiae*" but as far as I know, this is the species that is the most heavily studied in terms of transcriptome, proteome and ribosomal profiling to look for de novo genes. Very little (almost none) of the previous data is integrated here so that we do not know how the current results compare with previous studies and what novelties emerge from these analyses. The focal genome for the analysis of de novo genes *S. cerevisiae* and the data in the other species is not that much exploited. Some of the de novo genes uncovered by the analysis are actually already described and annotated in the yeast genome (e.g. AUA1) so it is not clear what new we are learning. Also, quite a bit of work has been done on antisense transcription in yeast and there is very little of this described here. I believe that the observation that de novo genes emerge from antisense transcripts is interesting and deserve further attention but as presented now, this would not be examined in enough details so that it would provide broad insights that warrant publication in a generalist journal such as this one.

Reviewer #2:

Remarks to the Author:

Using deep RNA-seq, ribosomal profiling and stringent filtering pipeline, this study identified hundreds of de novo genes in the baker's yeast (*Saccharomyces cerevisiae*) in the phylogenetic context of 11 yeast species, which has very compact genome compared to other higher organisms. Further analyses found that about half of these de novo genes originated from the antisense strands of pre-existing old genes, and about 1/3 are likely to have arisen from bidirectional promoters. Despite that a number of studies on identification of de novo genes have been reported in other organisms recently, the results expanded knowledge about the mechanism and process of de novo gene origination.

Given that the whole-length cDNA sequencing technology using the PacBio platform has been quite mature now, the current study is highly expected to apply this technology to significantly improve the identification and annotation of novel transcripts at least in *S. cerevisiae*. This will insure the discovery in this study to reach much higher level and endure longer life-span.

In addition, the current Discussions are too concise and thus the evolutionary significances of discoveries were not fully expressed. For example, when discussing the antisense transcripts, the authors didn't discuss its significance in compact genomes compared to other genomes with more "junk" DNAs. Neither did they deeply dig the observation that "Although it has been previously reported that overlapping antisense transcripts often represses the expression of the sense transcript (Huber et al. 2016), we observed a strong positive correlation between the changes in expression of the two transcripts across conditions." The co-regulation of yeast de novo genes at the protein level

encoded by sense and antisense strands has been reported in the paper of Li et al (2010), which was also cited by the authors. This may explain the unusual strong positive correlation better than simply attributing to passive turnover of de novo genes. Future perspectives are also lack in the Discussion. The paper was well written and I only caught few typos:
Line 61, represented11%  represented 11%
Line 271, de novo should be italic.

Reviewer #3:
Remarks to the Author:
General comments

The formation of new transcripts could be an important early step in the process of de novo gene birth. To better understand this process in an organism with a gene-dense genome, Blevins et al. attempt to identify a comprehensive set of de novo transcripts (i.e., transcripts derived from previously untranscribed sequence) in *S. cerevisiae*. They do so by analyzing transcriptomes and translatoe data across diverse yeast species, data which they had published previously but did not analyze in this light yet (Belvins 2019 a and b). They identify many transcripts in the focus species which are not detected in other species. They find that many of these putative de novo transcripts are translated and that many are located on the opposite strand of conserved genes.

While I find the topic of the study very interesting, the manuscript would greatly improve if the authors better introduced the literature context and highlighted more clearly what is novel about their methods or findings. For instance, the idea of comparing transcripts across closely related species was implemented previously in the context of de novo gene birth (eg Zhao et al 2014) – is anything novel about the authors approach besides its application to yeast? The fact that de novo gene birth is prevalent in yeast has been established by many publications (eg Carvunis et al 2012, Lu et al 2017, Vakirlis et al 2017, Wu et al 2018, Nielly-Thibault et al 2019), why do the authors not present this context and write that “little is known” about it? Similarly the fact many unannotated *cerevisiae* transcripts present signatures of translation is also well established (including through work by some of the authors of this study among many others), and should be acknowledged as such. The observation that de novo genes in yeast tend to overlap other genes was also made before (Carvunis et al 2012, Wu et al 2018; also the well described example MDF1 described in Li et al 2010 and Li et al 2014 which is not cited), as was that of the possible role of bidirectional promoters (Vakirlis et al 2017). The question of how novel transcripts emerge in a compact genome relative to a large genome is very interesting, but I cannot find a comparative analysis, or even a discussion of what the conclusions of the authors are regarding this question.

Furthermore, there are caveats in the approach that reduce enthusiasm First, the use of BLAST has fallen out of favor in the field because the mere absence of homology is consistent with many scenarios other than de novo birth, including rapid divergence, incomplete genomic sequencing, horizontal transfer from a species without a sequenced genome, etc (it is also questionable why only 35 outgroups are used, rather than NR for instance). The authors implement apply another step besides BLAST: they map each transcript to syntenic regions among the species in the clade, and then eliminate all *S. cerevisiae* transcripts that overlap a transcript from a different species in this clade within the syntenic region, regardless of identified homology. This step may be designed to address the above problem, but the fundamental issue remains, which is that absence of identified homology is not sufficient to prove evolutionary novelty. Furthermore, transcription is context-dependent. We cannot eliminate the possibility that a transcript is sometimes expressed in a species by sequencing the transcriptome in only two conditions. Perhaps the conditions under which a sequence is transcribed has changed over evolutionary time? Perhaps the depth of sequencing was not enough to detect expression of this transcript reliably enough for the assembly to work? I think these issues make it very difficult to make strong claims about de novo transcripts. Ideally, the authors may

consider the robust methods that have been developed to demonstrate the de novo origin of ORFs (based on syntenic alignments) and consider what the proper analogue is for transcripts. This would be a major advance. That said, if this is too ambitious, these major caveats should at least be discussed in the manuscript. It would strengthen the work to clearly lay out what the strengths and limitations of their approach are, and might also help clarify the novelty.

Some of the technical aspects of the analyses were lacking in the methods (or I could not find them, if so I apologize). For instance, what is the reference annotation used by the authors (there are several possible sources of varying completeness)? What parameters are used to define synteny blocks? Providing your in house scripts in a data repository would greatly help reviewing, and further guarantee reproducibility of the analyses.

Additional comments

- There are several numerical confusions throughout the manuscript, such as mention of 99 putative novel peptides in the discussion, when the result section describes 144, or of only 2270 verified ORFs in the caption of supp fig. 7 rather than ~5000?. Please verify and clarify.
- What is the estimated false positive rate of RibORF for low expression transcripts?
- The claim that de novo and genus specific genes are enriched in stress conditions is problematic, as it seems clear from Fig 1g that they are also enriched in the normal condition. Could it be that they appear condition specific simply because they are expressed at lower levels, and therefore difficult to detect (in a way amplified in the stress condition because of ribosomal transcripts as commented by the authors)?
- The term "novel" is used ambiguously in a few places throughout the manuscript; it could refer to evolutionary novelty or it could be a synonym for "unannotated." Similarly, there are times when the relationship between "unannotated" and "uncharacterized" induces confusion since ~1000 loci are annotated as uncharacterized in the saccharomyces genome database. Please remove these ambiguities from the manuscript.
- In a number of places in the manuscript there is confusion between de novo transcripts and de novo genes. Some of the authors' own previous work (e.g. Ruiz-Orera 2018) indicate that many de novo coding sequences appear to be evolving neutrally and show no evidence of selected function. If the authors want to use a definition of "gene" that does not imply selected function they should make this very clear, or use alternative terminology.
- p4Line 61: there should be a space between 'represented' and '11%'
- p5Line 110: I think you meant 'the number of transcripts' instead of 'number of genes' as you categorized the transcripts and we don't know whether all are genes or not. This also applies to the following paragraph
- p6.Figure 1: I think panel e and f shares the color legend but it was plotted only for panel f
- p7.Line 130: 'g' representing panel g should be bold.
- P. 7, Line 137 to 144: The number 70% refers to the genome overall (12MB) but should be divided by two if you consider both strands. The abundance of functional sequences like regulatory regions doesn't necessarily hinder the transcription of non-coding genomic elements. Please reconsider this reasoning to introduce your analyses
- p8.Line 161: each other should be written separately
- p9, last paragraph: There seems to be a problem with Huber et al 2009 reference, the first author is listed as Zhenyu Xu on journal's website; In that paper size of a single NFR is taken as 131nt. ~400nt looks like the distance between 2 divergent transcription start sites. So please check.
- P. 10, first paragraph, it seems like Fig 3a is not referenced at the right sentence, and I could not see a figure showing the AUG and NUG results.
- P10, line 219, did the authors mean MIN3 rather than MINI?
- p11.Figure 3b:
 - There is a typo on the second line, I think it should be annotated
 - For unannotated it would be clearer it is was written 'translated' rather than only in the caption

- p12.Line 240 to 251:
 - For d: How important it is to write mean and median here? The plot already shows one of them and you clearly show the significance level. Also the numbers on the plot is aa while you mention nucleotide here, which is confusing.
 - For e and f: how you calculate coding score or isoelectric point needs to be in the methods section, which I don't see now. Since it also says previously developed for cipher, a reference is needed. Also 'Peptides' has a paper published, so you need to cite it.
- p13.Line 310: there should be space between 'emerge' and 'from'

Blevins et al. Uncovering *de novo* gene birth in yeast using deep transcriptomics.

Response to reviewers

Reviewer #1 (Remarks to the Author):

The authors perform a RNAseq experiment on 11 yeast species spanning billions of years of evolution. They use these RNAseq experiments (1 replicate per species) and ribosomal profiling experiments (in *S. cerevisiae* only) to identify putative *de novo* genes. They focus their detailed analyses on *S. cerevisiae*. They find that antisense transcripts (on opposite strands from a coding sequence) may often contain *de novo* genes. There is no functional follow up on any of the findings to confirm expression and translation.

The work has been well performed and will provide a useful resource for the community and a very rich dataset to mine in the future. I do not see noticeable limitations to the methods and the analysis. However, the work is very descriptive and brings limited new findings to the field. The manuscript appears not to be considering all of the previous work that has been done on yeast *de novo* genes. This would have helped see how their work brings the field forward. They do cite the relevant papers but do not really compare their results with previous papers examining *de novo* genes and their properties so we do not see what are the novel observations that they made. In the abstract, the authors mention “However, little is known about the formation of new transcripts in more densely-packed genomes such as *Saccharomyces cerevisiae*’ but as far as I know, this is the species that is the most heavily studied in terms of transcriptome, proteome and ribosomal profiling to look for *de novo* genes. Very little (almost none) of the previous data is integrated here so that we do not know how the current results compare with previous studies and what novelties emerge from these analyses. The focal genome for the analysis of *de novo* genes *S. cerevisiae* and the data in the other species is not that much exploited. Some of the *de novo* genes uncovered by the analysis are actually already described and annotated in the yeast genome (e.g. AUA1) so it is not clear what new we are learning. Also, quite a bit of work has been done on antisense transcription in yeast and there is very little of this described here. I believe that the observation that *de novo* genes emerge from antisense transcripts is interesting and deserve further attention but as presented now, this would not be examined in enough details so that it would provide broad insights that warrant publication in a generalist journal such as this one.

We appreciate the comments of the reviewer; they have been very useful to improve the paper and given us an opportunity to emphasize the main novelties and the differences with previous papers. We provide a point-by-point response below:

Comment 1: “However, little is known about the formation of new transcripts in more densely-packed genomes such as *Saccharomyces cerevisiae*” but as far as I know, this is the species that is the most heavily studied in terms of transcriptome, proteome and ribosomal profiling to look for *de novo* genes.

We have extended this sentence in the Abstract to put it into context and clarify that most previous studies focused on ORFs and not transcripts: “However, little is known about how new transcripts are formed in more densely-packed genomes such as *Saccharomyces cerevisiae*, as the existing genome-wide studies have focused on genomic open reading frames (ORFs) and not transcripts.” More than half of the events we describe correspond to non-coding transcripts, which would not have been detected in ORF-based studies.

Comment 2: Very little (almost none) of the previous data is integrated here so that we do not know how the current results compare with previous studies and what novelties emerge from these analyses.

Our approach is very different from the ones employed in previous studies which focused on ORFs and/or did not use transcriptomics data from other species. As a result, we are uncovering many *de novo* transcripts that were not previously detected, including both coding and non-coding transcripts. We also perform a detailed analysis of the genomic location of the transcripts and the changes in gene expression levels in normal and oxidative stress conditions. We establish that half *de novo* transcripts are overlapping other genes in antisense orientation and that many of these transcripts contain ORFs that are translated. Below we explain the changes made to the manuscript in more detail.

We have added a complete new section and figure (section “Comparison with other approaches”, pages 8 - 10 and Figure 3) that compares our strategy to other genome-wide studies in *S. cerevisiae* (Carvunis et al., 2012; Vakirlis et al., 2017; Wu et al., 2018). In this section we compare the outcome of running the pipeline to identify *de novo* transcripts under different scenarios – 1. transcriptomics and annotations in all species (our study), 2. transcriptomics in *S. cerevisiae* only (akin to Carvunis) and, 3. Annotations in all species (akin to Vakirlis)-. We find that strategies 2. and 3. can result in a substantial overestimation of the number of *de novo* genes and, in case 3., also miss the majority of cases because they are not annotated. In the same Figure we also present a new table in which we compare the specific *de novo* proteins identified in the different studies, and which clearly shows that the majority of *de novo* proteins identified in the present study are new.

Comment 3: The focal genome for the analysis of *de novo* genes *S. cerevisiae* and the data in the other species is not that much exploited.

We have focused on *S. cerevisiae* because we could generate ribosome profiling data for this species in exactly the same conditions as for the RNA-Seq data. We feel any data we may present for the other species will necessarily be more incomplete and will thus would not add much to the conclusions of the study. In any case we have made the RNA-Seq data of all species available in case other researchers wish to explore it further.

Comment 4: Some of the *de novo* genes uncovered by the analysis are actually already described and annotated in the yeast genome (e.g. AUA1) so it is not clear what new we are learning.

Using RNA-Seq data and Ribo-Seq we uncovered 54 *de novo* proteins from *S. cerevisiae* that are not annotated. Several novel examples are now shown in new Figure 6 and the new supplementary Table 7; this includes novel *bona fide* translated ORFs that overlap the NADP+ dependent arabinose dehydrogenase gene and the CTR3 copper transporter gene. In addition, we also discovered a number of translated *de novo* ORFs in transcripts previously believed to be non-coding, such as Ruf5-1 and Ruf5-2, which overlap the CUP1 methallonein gene and which are overexpressed under stress. These examples illustrate the power of our approach to discover not yet described *de novo* genes.

Comment 5: Also, quite a bit of work has been done on antisense transcription in yeast and there is very little of this described here. I believe that the observation that *de novo* genes emerge from antisense transcripts is interesting and deserve further attention but as

presented now, this would not be examined in enough details so that it would provide broad insights that warrant publication in a generalist journal such as this one.

We agree that this is a novel and interesting aspect of our work. To our knowledge, this study is the first to show that antisense transcripts are strongly over-represented among *de novo* genes in yeast (Figure 5). In addition, the strong correlation in expression level changes between sense and antisense transcripts suggests that these *de novo* genes are likely to participate in similar cellular processes as the overlapped gene. We have extended the discussion of these findings: “Some antisense transcripts may play a role in controlling the abundance of the protein encoded by the sense gene (Camblong et al. 2007; Pelechano and Steinmetz, 2013). Huber et al. 2016 repressed antisense transcripts of 162 yeast genes and observed an effect in about 25% of the genes, mostly a weak decrease in the amount of the sense protein (Huber et al., 2016). Here we observed that changes in the expression of sense and antisense genes tended to be positively correlated. On the basis of this, and on specific observations for gene pairs with experimental information for both members of the pair (AUA1-WWM1, VAM10-VPS5 and MDF1-ADF1), we can speculate that the functions of *de novo* proteins in antisense transcripts may often be related to that of the overlapped gene, by being involved in related cellular processes or by regulating the activity of the gene”. (pages 487-497).

Reviewer #2 (Remarks to the Author):

Using deep RNA-seq, ribosomal profiling and stringent filtering pipeline, this study identified hundreds of de novo genes in the baker's yeast (*Saccharomyces cerevisiae*) in the phylogenetic context of 11 yeast species, which has very compact genome compared to other higher organisms. Further analyses found that about half of these de novo genes originated from the antisense strands of pre-existing old genes, and about 1/3 are likely to have arisen from bidirectional promoters. Despite that a number of studies on identification of de novo genes have been reported in other organisms recently, the results expanded knowledge about the mechanism and process of de novo gene origination.

Given that the whole-length cDNA sequencing technology using the PacBio platform has been quite mature now, the current study is highly expected to apply this technology to significantly improve the identification and annotation of novel transcripts at least in *S. cerevisiae*. This will insure the discovery in this study to reach much higher level and endure longer life-span.

In addition, the current Discussions are too concise and thus the evolutionary significances of discoveries were not fully expressed. For example, when discussing the antisense transcripts, the authors didn't discuss its significance in compact genomes compared to other genomes with more "junk" DNAs. Neither did they deeply dig the observation that "Although it has been previously reported that overlapping antisense transcripts often represses the expression of the sense transcript (Huber et al. 2016), we observed a strong positive correlation between the changes in expression of the two transcripts across conditions." The co-regulation of yeast de novo genes at the protein level encoded by sense and antisense strands has been reported in the paper of Li et al (2010), which was also cited by the authors. This may explain the unusual strong positive correlation better than simply attributing to passive turnover of de novo genes. Future perspectives are also lack in the

Discussion.

The paper was well written and I only caught few typos:

Line 61, represented 11%  represented 11%

Line 271, de novo should be italic.

We appreciate the comments of the reviewer. We have extensively revised our manuscript to include the different aspects mentioned in the review. Below we explain these additions in more detail, and provide a point-by-point response to the reviewer's comments.

Comment 1: Given that the whole-length cDNA sequencing technology using the PacBio platform has been quite mature now, the current study is highly expected to apply this technology to significantly improve the identification and annotation of novel transcripts at least in *S. cerevisiae*.

Although we agree that using PacBio to obtain whole-length cDNA sequencing is very effective to obtain full-length transcripts, the cost/benefit relationship of this strategy at this point is not so clear, especially considering that we already implemented many controls to ensure that the transcripts we obtained were as complete as possible, and that the complexity of the yeast genome is far less than that of multicellular eukaryotes like humans.

Firstly, we performed very deep polyA+ RNA sequencing, with about 60 million reads per species. This means we had very high coverage even for lowly expressed genes, which greatly facilitates the reconstruction of the transcripts. Secondly, and perhaps even more importantly, we used spike-ins (a mixture of known mRNAs at different concentrations) as a control for the transcript assembly process. As shown in Supplementary Figure 7, this control allowed us to determine the minimum mRNA concentration or expression value (in

transcripts per million units, or TPM) required for the complete assembly of an mRNA species in the sample. We established that the vast majority of transcripts (> 95%) expressed above 15 TPM could be fully recovered in our assembly pipeline. Consequently, we used this expression cut-off throughout the study, which ensured that the transcripts we analysed were complete.

Comment 2: In addition, the current Discussions are too concise and thus the evolutionary significances of discoveries were not fully expressed. For example, when discussing the antisense transcripts, the authors didn't discuss its significance in compact genomes compared to other genomes with more "junk" DNAs.

We have significantly expanded the Results and Discussion sections to include these and other aspects mentioned by the reviewers. We believe that manuscript is now much more informative and the significance of the discoveries is more clear.

Regarding the antisense transcripts, we now provide more examples in new Figure 6 and new supplementary Table 7 and include additional text in the corresponding section (pages 15-16). We have rewritten the corresponding section in the Discussion as follows:

"Some antisense transcripts may play a role in controlling the abundance of the protein encoded by the sense gene (Camblong et al. 2007; Pelechano and Steinmetz, 2013). Huber et al. 2016 repressed antisense transcripts of 162 yeast genes and observed an effect in about 25% of the genes, mostly a weak decrease in the amount of the sense protein (Huber et al., 2016). Here we observed that changes in the expression of sense and antisense genes tended to be positively correlated. On the basis of this, and on specific observations for gene pairs with experimental information for both members of the pair (AUA1-WWM1, VAM10-VPS5 and MDF1-ADF1), we can speculate that the functions of *de novo* proteins in antisense transcripts may often be related to that of the overlapped gene, by being involved in related cellular processes or by regulating the activity of the gene. An interesting example was a *de novo* ORF encoding a protein of 64 amino acids that overlapped CUP-1, a metallothionein-encoding gene. The expression of the two transcripts of the sense-antisense pair increased about two fold under oxidative stress conditions (from 246-304 to 570-700 TPM), suggesting that both proteins have a role in the response to stress." (lines 487-499).

We have also included a comparison on the prevalence and genomic context of *de novo* antisense transcripts between yeast and human:

"The use of transcriptomes from multiple species was previously used to investigate *de novo* gene evolution in *Drosophila* (Zhao et al. 2014), primates (Ruiz-Orera et al., 2015) and rice (Zhang et al., 2019), but not in the model unicellular eukaryote, *S. cerevisiae*. We wanted to investigate how the compactness of the yeast genome, with 70% of the sequence covered by coding sequences, would impact the formation of new transcripts. We found a very strong enrichment in transcripts that overlapped other exons in the opposite strand; this type of transcripts represented 50% of the *de novo* transcripts in yeast but only 10% of the *de novo* transcripts in humans (Ruiz-Orera et al., 2015)." (lines 389-396).

Comment 3: Neither did they deeply dig the observation that "Although it has been previously reported that overlapping antisense transcripts often represses the expression of the sense transcript (Huber et al. 2016), we observed a strong positive correlation between the changes in expression of the two transcripts across conditions." The co-regulation of yeast *de novo* genes at the protein level encoded by sense and antisense strands has been reported in the

paper of Li et al (2010), which was also cited by the authors. This may explain the unusual strong positive correlation better than simply attributing to passive turnover of *de novo* genes.

We have examined in detail the available information about MDF1, which is first reported in Li et al. (2010) and further investigated in Li et al. (2014). We now dedicate a full paragraph to expand on this example: “The only previously well-described example of a *de novo* gene overlapping another gene in the opposite strand in *S. cerevisiae* is MDF1, which overlaps ADF1 (Li et al., 2010; Li et al., 2014). MDF1 has been proposed to promote vegetative growth and is negatively regulated by the product of ADF1. Although previously characterized as likely to be *S. cerevisiae*-specific, our transcriptomics-based approach classified MDF1 as genus-specific (Figure 6). This does not preclude a *de novo* origin of the gene, it simply extends the homology to more distant species which expressed related transcripts.” (lines 338-343).

In the new Figure 6 we report 5 additional examples of antisense *de novo* transcripts with evidence of protein translation. This includes AUA1/WWM1 and VAM10/VPS5 but also newly discovered translated ORFs that overlap CUP1, ARA1 and CTR3. We mention in the Discussion that these *de novo* proteins could be involved in related cellular processes (following co-expression in stress versus normal) or regulate the activity of the sense gene.

Comment 4: Future perspectives are also lack in the Discussion.

We have added the following paragraph at the end of the Discussion:

“This work establishes, using transcriptomics data from multiple species and genomic synteny, that about 5% of the transcripts in yeast have arisen *de novo* in recent times. Most of these transcripts are not present in the annotations, probably because the encoded proteins are smaller than 100 amino acids, and thus they have remained hidden. We have found that a disproportionately large fraction of them is overlapping other genes in the opposite strand and propose that this configuration can enhance the functionalization of the new transcripts, which may inherit regulatory features of the overlapped gene. As this configuration is not so common in more conserved transcripts, this may especially serve relative short-time scale adaptations (in the order of tens of Millions of years). Much work still needs to be done to understand the advantages these transcripts, and the proteins they encode, may have provided to the organism. Large-scale experimental transcript inactivation screenings coupled with the monitoring of gene expression changes may provide new clues to their possible regulatory activities or their involvement in increased organism survival in the face of environmental challenges.” (lines 502-513).

Comment 5: The paper was well written and I only caught few typos:

Line 61, represented11%  represented 11%

Line 271, *de novo* should be italic.

We have corrected these typos.

Reviewer #3 (Remarks to the Author):

General comments

The formation of new transcripts could be an important early step in the process of de novo gene birth. To better understand this process in an organism with a gene-dense genome, Blevins et al. attempt to identify a comprehensive set of de novo transcripts (i.e., transcripts derived from previously untranscribed sequence) in *S. cerevisiae*. They do so by analyzing transcriptomes and translome data across diverse yeast species, data which they had published previously but did not analyze in this light yet (Blevins 2019 a and b). They identify many transcripts in the focus species which are not detected in other species. They find that many of these putative de novo transcripts are translated and that many are located on the opposite strand of conserved genes.

While I find the topic of the study very interesting, the manuscript would greatly improve if the authors better introduced the literature context and highlighted more clearly what is novel about their methods or findings. For instance, the idea of comparing transcripts across closely related species was implemented previously in the context of de novo gene birth (eg Zhao et al 2014) – is anything novel about the authors approach besides its application to yeast? The fact that de novo gene birth is prevalent in yeast has been established by many publications (eg Carvunis et al 2012, Lu et al 2017, Vakirlis et al 2017, Wu et al 2018, Nielly-Thibault et al 2019), why do the authors not present this context and write that “little is known” about it? Similarly the fact many unannotated *cerevisiae* transcripts present signatures of translation is also well established (including through work by some of the authors of this study among many others), and should be acknowledged as such. The observation that de novo genes in yeast tend to overlap other genes was also made before (Carvunis et al 2012, Wu et al 2018; also the well described example MDF1 described in Li et al 2010 and Li et al 2014 which is not cited), as was that of the possible role of bidirectional promoters (Vakirlis et al 2017). The question of how novel transcripts emerge in a compact genome relative to a large genome is very interesting, but I cannot find a comparative analysis, or even a discussion of what the conclusions of the authors are regarding this question.

Furthermore, there are caveats in the approach that reduce enthusiasm. First, the use of BLAST has fallen out of favor in the field because the mere absence of homology is consistent with many scenarios other than de novo birth, including rapid divergence, incomplete genomic sequencing, horizontal transfer from a species without a sequenced genome, etc (it is also questionable why only 35 outgroups are used, rather than NR for instance). The authors implement another step besides BLAST: they map each transcript to syntenic regions among the species in the clade, and then eliminate all *S. cerevisiae* transcripts that overlap a transcript from a different species in this clade within the syntenic region, regardless of identified homology. This step may be designed to address the above problem, but the fundamental issue remains, which is that absence of identified homology is not sufficient to prove evolutionary novelty. Furthermore, transcription is context-dependent. We cannot eliminate the possibility that a transcript is sometimes expressed in a species by sequencing the transcriptome in only two conditions. Perhaps the conditions under which a sequence is transcribed has changed over evolutionary time? Perhaps the depth of sequencing was not enough to detect expression of this transcript reliably enough for the assembly to work? I think these issues make it very difficult to make strong claims about de novo transcripts. Ideally, the authors may consider the robust methods that have been developed to demonstrate the de novo origin of ORFs (based on syntenic alignments) and consider what the proper analogue is

for transcripts. This would be a major advance. That said, if this is too ambitious, these major caveats should at least be discussed in the manuscript. It would strengthen the work to clearly lay out what the strengths and limitations of their approach are, and might also help clarify the novelty.

Some of the technical aspects of the analyses were lacking in the methods (or I could not find them, if so I apologize). For instance, what is the reference annotation used by the authors (there are several possible sources of varying completeness)? What parameters are used to define synteny blocks? Providing your in house scripts in a data repository would greatly help reviewing, and further guarantee reproducibility of the analyses.

We would like to thank the reviewer for their relevant comments and attention to detail, which has greatly helped us to improve the manuscript. We have separated out the different comments of the reviewer, and included our responses to each comment. We have also included responses to the additional comments at the end of this document.

Comment 1: For instance, the idea of comparing transcripts across closely related species was implemented previously in the context of de novo gene birth (eg Zhao et al 2014) – is anything novel about the authors approach besides its application to yeast?

Although this approach has been used in Drosophila (Zhao et al., 2014), primates (Ruiz-Orera et al., 2015) and rice (Zhang et al., 2019), there was a gap in the use of similar methodologies in yeast. We explain this in the manuscript as follows:

“The use of transcriptomes from multiple species was previously used to investigate de novo gene evolution in Drosophila (Zhao et al. 2104), primates (Ruiz-Orera et al., 2015) and rice (Zhang et al., 2019), but not in the model unicellular eukaryote, *S. cerevisiae*. We wanted to investigate how the compactness of the yeast genome, with 70% of the sequence covered by coding sequences, would impact the formation of new transcripts. We found a very strong enrichment in transcripts that overlapped other exons in the opposite strand; this type of transcripts represented 50% of the *de novo* transcripts in yeast but only 10% of the *de novo* transcripts in humans (Ruiz-Orera et al., 2015).” (lines 389-396).

Comment 2: The fact that de novo gene birth is prevalent in yeast has been established by many publications (eg Carvunis et al 2012, Lu et al 2017, Vakirlis et al 2017, Wu et al 2018, Nielly-Thibault et al 2019), why do the authors not present this context and write that “little is known” about it?

We have extended this sentence in the Abstract to put it into context and clarify that most previous studies focused on ORFs and not transcripts: “However, little is known about how new transcripts are formed in more densely-packed genomes such as *Saccharomyces cerevisiae*, as the existing genome-wide studies have focused on genomic open reading frames (ORFs) and not transcripts.”

Comment 3: Similarly the fact many unannotated *cerevisiae* transcripts present signatures of translation is also well established (including through work by some of the authors of this study among many others), and should be acknowledged as such.

We agree that many studies have shown that many unannotated transcripts are indeed translated. However, methods based on the three nucleotide periodicity of the ribosome profiling reads had not been previously applied to studies of *de novo* genes in yeast. These

methods can distinguish between *bona fide* translation events and other types of signals caused by non-translational ribosome scanning or ribonucleoprotein particles.

Comment 4: The observation that *de novo* genes in yeast tend to overlap other genes was also made before (Carvunis et al 2012, Wu et al 2018; also the well described example MDF1 described in Li et al 2010 and Li et al 2014 which is not cited), as was that of the possible role of bidirectional promoters (Vakirlis et al 2017).

We have made a substantial revision of the paper to address these questions; we summarize below the main additions:

Firstly, we have added a complete new section and figure (section “Comparison with other approaches”, pages 9 - 10 and Figure 3) that compares different strategies and studies, including Carvunis et al., 2012; Vakirlis et al., 2017 and Wu et al., 2018.

In this section we clarify what is novel about our strategy relative to previous studies: “Our methodology to find *de novo* genes in *S. cerevisiae* was different to previous approaches, because, in addition to annotated genes from multiple species, our study also included thousands of *de novo* assembled transcripts from 11 yeast species. The inclusion of these unannotated transcripts serves to minimize the erroneous classification of transcripts as *de novo* (false positives), caused by a failure to detect the homologues in the other species, and at the same time to be as sensitive as possible by including transcripts that are not present in the annotations.” Then we run the same pipeline but only considering transcriptomics data from *S. cerevisiae* (akin to Carvunis et al., 2012), or only annotated genes for all the species (akin to Vakirlis et al., 2018). In both cases we observe an excess of transcript classified as *de novo*, suggesting that the previously used strategies could include many misclassified genes. Finally, we compare the set of *de novo* genes obtained in different studies, focusing on *S. cerevisiae*-specific annotated protein-coding genes, as this provides a common denominator for all studies considered. As expected, we found that these lists are quite different but that they also include a substantial number of common *de novo* genes.

Secondly, we have generated a new section and figure on *de novo* genes that overlap other genes, “Examples of *de novo* proteins in sense-antisense pairs” (pages 15-16). This includes the MDF1 example but also a new paragraph dedicated to *de novo* ORFs with evidence of translation identified in our study, such as those overlapping CUP1, ARA1 and CTR3 (see below).

Lines 338-343: “The only previously well-described example of a *de novo* gene overlapping another gene in the opposite strand in *S. cerevisiae* is MDF1, which overlaps ADF1 (Li et al., 2010; Li et al., 2014). MDF1 has been proposed to promote vegetative growth and is negatively regulated by the product of ADF1. Although previously characterized as likely to be *S. cerevisiae*-specific, our transcriptomics-based approach classified MDF1 as genus-specific (Figure 6). This does not preclude a *de novo* origin of the gene, it simply extends the homology to more distant species which expressed related transcripts.”

Lines 361-368: “We identified other *de novo* proteins that have not been described in the literature and remain unannotated (Figure 6, more details of these and other examples can be found in Supplementary Table 7). For instance, we observed two translated ORFs encoding 64 amino acids long proteins in the opposite orientation to the two CUP1 gene copies located in chromosome 8. CUP1 encodes a metallothionein, which mediates resistance to high concentrations of copper and cadmium (Fogel and Welch, 1982). Interestingly, the origin of CUP1 is also quite recent; our pipeline classified this gene as

genus-specific. Both CUP1 and the newly discovered antisense ORFs were strongly over-expressed in oxidative stress conditions, suggesting that both could be involved in the response to oxidative stress.”

Lines 370-378: “Another example was an ORF encoding a 54 amino acid protein overlapping ARA1 on the opposite orientation. ARA1 encodes a NADP⁺ dependent arabinose dehydrogenase and a deficient mutant showed increased susceptibility to H₂O₂-induced stress (Amako et al., 2006). In line with this, we found that the expression of ARA1 was about two fold in oxidative stress conditions than in normal growth conditions. A very similar pattern was observed for the new overlapping *de novo* protein, suggesting that this protein could be involved in the response to stress too. A third example was a novel protein of 51 amino acids encoded by a recently originated transcript overlapping the copper transporter CTR3 (Pena et al., 2000). In this case both proteins were well-expressed in rich medium but showed only residual expression (TPM < 15) in oxidative stress conditions.”

Regarding the role of bidirectional promoters, we found that this mechanism is supported for a number of *de novo* genes, but the relative frequency of these cases in *de novo* transcripts is similar to the frequency in conserved genes (Figure 5). This is in sharp contrast to what was reported in Vakirlis et al. 2018, which found an over-representation of this configuration in *de novo* genes.

Comment 5: The question of how novel transcripts emerge in a compact genome relative to a large genome is very interesting, but I cannot find a comparative analysis, or even a discussion of what the conclusions of the authors are regarding this question.

We have compared the frequency of yeast *de novo* transcripts which are in an overlapping orientation to transcripts in the same orientation in humans and the difference is striking; whereas ~50% of *de novo* transcripts have antisense overlap in yeast, it is only ~10% in the human genome. We find that in the compact yeast genome the alternative strand is used more frequently for the creation of new transcripts than in other organisms with a less-dense genome. We have included this comparison in the paper: “We found a very strong enrichment in transcripts that overlapped other exons in the opposite strand; this type of transcripts represented 50% of the *de novo* transcripts in yeast but only 10% of the *de novo* transcripts in humans (Ruiz-Orera et al., 2015).” (lines 394-397).

Comment 6: Furthermore, there are caveats in the approach that reduce enthusiasm. First, the use of BLAST has fallen out of favor in the field because the mere absence of homology is consistent with many scenarios other than *de novo* birth, including rapid divergence, incomplete genomic sequencing, horizontal transfer from a species without a sequenced genome, etc (it is also questionable why only 35 outgroups are used, rather than NR for instance). The authors implement another step besides BLAST: they map each transcript to syntenic regions among the species in the clade, and then eliminate all *S. cerevisiae* transcripts that overlap a transcript from a different species in this clade within the syntenic region, regardless of identified homology. This step may be designed to address the above problem, but the fundamental issue remains, which is that absence of identified homology is not sufficient to prove evolutionary novelty. Perhaps the conditions under which a sequence is transcribed has changed over evolutionary time? Perhaps the depth of sequencing was not enough to detect expression of this transcript reliably enough for the assembly to work? I think these issues make it very difficult to make strong claims about *de novo* transcripts. Ideally, the authors may consider the robust methods that have been developed to demonstrate the *de novo* origin of ORFs (based on syntenic alignments) and consider what the proper analogue is for transcripts. This would be a major advance. That said, if this is too ambitious, these major

caveats should at least be discussed in the manuscript. It would strengthen the work to clearly lay out what the strengths and limitations of their approach are, and might also help clarify the novelty.

Based on our previous experience (e.g. Ruiz-Orera et al., 2015; Villanueva-Cañas et al., 2017) 35 outgroup species is sufficient to detect distant homologues and the results are easier to interpret than using nr (which includes many species with incomplete genomes). As noted by the reviewer, we also used genomic synteny in addition to BLAST searches to account for cases which may have gone undetected by BLAST. If there were any transcripts expressed in the corresponding syntenic genome region, we considered them to be putative homologues. Finally, whereas we only considered transcripts expressed at TPM > 15 in the focal species, all transcripts were included for the other species, regardless of expression level, increasing our ability to detect homologues.

We extensively discuss the caveats mentioned by the reviewer in two new paragraphs:

Lines 420-430: “Despite our conservative criteria to identify recently evolved *de novo* transcripts in *S. cerevisiae*, the possibility exists that a fraction of them have a more distant origin than the one we inferred. Rapid sequence divergence can make the detection of homologues difficult and result in an underestimation of the age of some genes (Albà and Castresana, 2005; Elhaik et al., 2016; Albà and Castresana, 2007). However, sequence evolution simulations indicate that this should have only a minor effect in comparisons of closely related species such as those within the *Saccharomyces* genus (Domazet-Loso et al., 2017). In addition, synteny-based analytical studies have recently shown that most genes for which we fail to detect homologues in more distant species, or orphans, are likely to have arisen *de novo* (Vakirlis et al., 2020). Perhaps more importantly, here we used genomic synteny in addition to sequence similarity searches across the transcripts, which should reduce even further the number of possible false positives.”

Lines 432-445: “Another possible source of errors in the identification of the branch of origin of a gene is the loss of the transcript in one or more species. Let’s imagine that a transcript originated in the common branch of the *Saccharomyces* genus and was subsequently lost in *S. bayanus* and *S. kudriazevii*, being currently only present in *S. cerevisiae*, *S. paradoxus* and *S. mikatae*. This transcript would have been classified as *de novo* in our pipeline, when it should be genus-specific. As genes of different age may be lost at different frequencies (Palmieri et al., 2014), it is difficult to estimate how often such losses may have happened. We dealt with this uncertainty by creating classes that were larger than a single internal branch and which grouped several branches and species; these classes were more robust to errors caused by secondary losses of genes, especially if this happened in a single species. Finally, we also have to consider that a transcript may completely change its expression pattern in one or more species, and become undetectable when using the same conditions for all species. This is probably relatively rare and the transcript may still maintain some basal expression levels in rich medium. In this regard, we observed that 95% of the *S. cerevisiae* annotated genes could be detected in rich medium (TPM > 2). We also have to consider that we used high sequencing coverage, which facilitates the detection of lowly expressed genes.”

Comment 7: Some of the technical aspects of the analyses were lacking in the methods (or I could not find them, if so I apologize). For instance, what is the reference annotation used by the authors (there are several possible sources of varying completeness)? What parameters are used to define synteny blocks? Providing your in house scripts in a data repository would greatly help reviewing, and further guarantee reproducibility of the analyses.

We have indicated that the *S. cerevisiae* reference genome is S288C (version 64-2-1), Genbank genome entry GCF_000146045.2. The synteny was inferred using maximal unique matches or MUMs. We have extended the corresponding section in Methods (“Genomic synteny comparisons”) to explain exactly how we determined if two transcripts were overlapping or not. Briefly, the synteny blocks are clusters of MUMs that are in close proximity (here we used < 100 nt). Then, for each transcript in a genome we first determine the synteny block it belongs to, using the genomic coordinates, and then check whether there is any transcript in the same region of the second genome. We have followed the recommendation of the reviewer and uploaded some of the code and processed data at https://github.com/willblev/Blevins_etal_2020.

Additional comments

- There are several numerical confusions throughout the manuscript, such as mention of 99 putative novel peptides in the discussion, when the result section describes 144, or of only 2270 verified ORFs in the caption of supp fig. 7 rather than ~ 5000 ?. Please verify and clarify.

We have revised these numbers and we detect 97 *de novo* transcripts with at least one translated ORF. As some of them contain more than one translated ORF this amounts to 123 putative novel peptides.

The low number of verified ORFs (2,270) was due to a masked error when importing a file into R that resulted in dropped rows. We have now corrected this and the number of verified ORFs we consider is 5068, of which 4932 were classified as coding (97.3%). We have corrected this in the manuscript and corresponding Supplementary Figure.

- What is the estimated false positive rate of RibORF for low expression transcripts?

We have previously estimated that the false positive rate of RibORF, using the same parameters as employed here (RibORF score > 0.7 , number Ribo-Seq reads ≥ 10), is 3.33 % (Ruiz-Orera et al., 2018, Nature Ecol Evol). This negative control was composed of ORFs in small nuclear and nucleolar RNAs from mouse that had mapped Ribo-Seq reads for reasons other than translation (presumably RNPs).

In *S. cerevisiae* we do not have enough data to build a similar negative control. However we can randomize the reads in the three different frames and see how the transcript expression level affects the results. We have done this for all transcripts, and for the subset of transcripts with TPM > 15 , that had ORFs with Ribo-Seq reads ≥ 10 . We have then re-calculated the RibORF score and counted the proportion of transcripts that have a score > 0.7 . Note that this test is conservative because we keep the original homogeneity value of the reads along the ORF. In the first group (all transcripts) 18.6% of the transcripts have score > 0.7 , whereas in the second group (TPM >15) this proportion is only slightly lower, 15.9%. This indicates that the method is relatively robust to differences in expression value. For the analysis of *de novo* genes and other conservation gene classes, we have focused on well-expressed genes (TPM > 15).

- The claim that *de novo* and genus specific genes are enriched in stress conditions is problematic, as it seems clear from Fig 1g that they are also enriched in the normal condition. Could it be that they appear condition specific simply because they are expressed at lower levels, and therefore difficult to detect (in a way amplified in the stress condition because of ribosomal transcripts as commented by the authors)?

We agree with the reviewer that the higher number of transcripts in the normal condition, which is observed for all conservation classes, is probably a consequence of the fact that the relative expression levels in stress conditions are more widely distributed, with a group of very abundant transcripts that mainly encode ribosomal proteins (as reported in Blevins et al., 2019b). Nevertheless when we compare the number of *de novo* and genus-specific transcripts that are only expressed during stress (taken together), with respect to the same number for the complete transcript set (expected), we can see a significant enrichment (6-8% observed *versus* 2.4% expected, p-value < 0.01 Fisher test). As mentioned in the Discussion, the enrichment is modest and merits further investigation.

- The term “novel” is used ambiguously in a few places throughout the manuscript; it could refer to evolutionary novelty or it could be a synonym for “unannotated.” Similarly, there are times when the relationship between “unannotated” and “uncharacterized” induces confusion since ~1000 loci are annotated as uncharacterized in the *Saccharomyces* genome database. Please remove these ambiguities from the manuscript.

We have tried to be very consistent in the revised version of the manuscript and use ‘novel’ to indicate ‘unannotated transcripts’ throughout the manuscript. We then use ‘characterized’ to refer to genes/proteins with a known function.

- In a number of places in the manuscript there is confusion between *de novo* transcripts and *de novo* genes. Some of the authors’ own previous work (e.g. Ruiz-Orera 2018) indicate that many *de novo* coding sequences appear to be evolving neutrally and show no evidence of selected function. If the authors want to use a definition of “gene” that does not imply selected function they should make this very clear, or use alternative terminology.

We apologize for the ambiguity and have changed the text to only include the term ‘*de novo* transcripts’, as this is the main subject of our study.

- p4Line 61: there should be a space between ‘represented’ and ‘11%’

We have corrected this error.

- p5Line 110: I think you meant ‘the number of transcripts’ instead of ‘number of genes’ as you categorized the transcripts and we don’t know whether all are genes or not. This also applies to the following paragraph

We have replaced ‘the number of genes’ by ‘the number of transcripts’.

- p6.Figure 1: I think panel e and f shares the color legend but it was plotted only for panel f

Thank you for your attention to detail; we have corrected this.

- p7.Line 130: ‘g’ representing panel g should be bold.

We have rectified this mistake.

- P. 7, Line 137 to 144: The number 70% refers to the genome overall (12MB) but should be divided by two if you consider both strands. The abundance of functional sequences like regulatory regions doesn't necessarily hinder the transcription of non-coding genomic elements. Please reconsider this reasoning to introduce your analyses

We have decided to maintain the original statement of '70% of the genome is spanned by annotated coding sequences' as we believe that most readers will interpret that this refers to coding sequences in either of the two strands, and dividing it by two could create confusion. We have eliminated the text referring to the regulatory regions, as indeed the presence of regulatory motifs does not necessarily hinder transcription of the region.

- p8.Line 161: each other should be written separately

We have corrected this typo.

- p9, last paragraph: There seems to be a problem with Huber et al 2009 reference, the first author is listed as Zhenyu Xu on journal's website; In that paper size of a single NFR is taken as 131nt. ~400nt looks like the distance between 2 divergent transcription start sites. So please check.

Indeed there was an error in the order of the authors, which we have now corrected. We would like to note that 400 bases is the maximum distance that we consider between the 5'ends of the transcripts. This distance could include the 5'UTR of one or both transcripts, as 5'UTR sequences are often missing from the annotations. According to Xu et al. two NRFs would require a minimum of 452 bases. So a maximum of 400 bases would seem an appropriate distance limit to identify transcripts that could share a promoter region. We have eliminated the sentence stating that the distance approximately corresponds to 1 NFR, which would be incorrect, and clarified that "these transcripts are likely to be separated by a single nucleosome free region (Huber et al. 2009)" (lines 291-292).

- P. 10, first paragraph, it seems like Fig 3a is not referenced at the right sentence, and I could not see a figure showing the AUG and NUG results.

Thank you for noticing this mistake. We are not considering NUG cases in the final set and thus have eliminated this sentence.

- P10, line 219, did the authors mean MIN3 rather than MINI?

Thank you again for your attention to detail. It is indeed MIN3 and we have corrected it.

- p11.Figure 3b:

- There is a typo on the second line, I think it should be annotated
- For unannotated it would be clearer it is was written 'translated' rather than only in the caption

We have made these corrections in the corresponding figure.

- p12.Line 240 to 251:

- For d: How important it is to write mean and median here? The plot already shows one of them and you clearly show the significance level. Also the numbers on the plot is aa while you mention nucleotide here, which is confusing.

- For e and f: how you calculate coding score or isoelectric point needs to be in the methods section, which I don't see now. Since it also says previously developed for cipher, a reference is needed. Also 'Peptides' has a paper published, so you need to cite it.

We have followed the recommendations of the reviewer and only show the median value; the unit of length is amino acids.

We have added a new section to methods, 'ORF properties' in which we describe how we calculate the coding score and isoelectric point, together with the appropriate references (lines 617-624).

- p13.Line 310: there should be space between 'emerge' and 'from'

We have fixed this typo.

Reviewers' Comments:

Reviewer #1:

Remarks to the Author:

The work is of excellent quality and will serve as a very valuable resource. I still believe that the novel overall insight gained from this paper is limited given everything that was done on the subject before even if it presents a rich resource for annotation and biological discovery. The authors argue that the novelty from previous studies is that they focus on transcripts rather than on ORFs in the genomic sequences. However, since it has been shown that most of the yeast genome is transcribed, the two approaches are not that different.

Reviewer #3:

Remarks to the Author:

I read the resubmitted manuscript by Blevins and colleagues with great interest. I appreciate the efforts that the authors made to include more of the literature, better describe caveats, and clarify that the goal of the study was to find novel transcripts rather than de novo coding elements. These additions greatly improve the manuscript.

A. Genes vs transcripts

I am not entirely satisfied with the rephrasing to "de novo transcripts". It is not pervasive throughout the manuscript, leading to considerable confusion (eg lines 125-130, 181, 197-204, 415...Figs2-3). Furthermore, some analyses and interpretations are used that would be appropriate for ORFs/genes but not for transcripts. When the authors attempt to compare their method with others, it is no longer relevant since the goals are so different (transcript vs ORF). It is not a question of being more or less "conservative". The authors should consider more deeply the interplay between ORF and transcript evolution when they want to make inferences about gene emergence. When describing the MDF1 example, they classify the "gene" as genus specific since a homologous transcript at the same locus exists in other species. However, as Li et al. show, frameshifts and stop codons mean that no comparable intact ORF exists in these species despite the nucleotide homology. The coding product itself is therefore species specific regardless of how old the transcript is. I do not know if the authors intend to be claiming otherwise—the paragraph is ambiguous on this point—but it should be made clear that this is not a contradiction of Li et al. More generally, in their analyses and discussions of de novo coding transcripts (statistics and examples), consideration should be given to ORF evolution as well. Otherwise, it is OK to write a paper about species-specific transcripts without invoking the de novo emergence of coding elements too.

B. Overlapping vs divergent

Regarding the finding that the species-specific transcripts tend to overlap annotated genes:

- What is the expectation? As the authors write, 70% of the genome is occupied by annotated genes. Given that the authors found novel transcripts, where would they be expected to be? Please provide simulations or statistical tests to support or refute the notion that novel transcripts are preferentially overlapping versus divergent.

- Please be clearer in your presentation of context. A number of papers have studied unannotated/non-coding/pervasive transcripts in cerevisiae, although without reference to de novo gene birth (e.g., CUTs, SUTs, XUTs). The idea that de novo gene emergence may occur frequently on antisense transcripts was also shown in the past, although not explicitly for species-specific transcripts. Both of these facts should be clear from the introduction.

- I hope the RNA sequencing protocol used was strand -specific, otherwise this finding is completely invalidated. The methods clarify it is so for ribo-seq, but nothing is written for RNA seq. I suppose this is only an oversight in the writing of the method section.

C. Additional comments

- I did not check in details, but please make sure that the phenotypes/ mechanisms described for the example ORFs in Figure 6 were not obtained by genetics. If they were, you could not distinguish whether the phenotypes are due to one or the other overlapping ORF, leading to an illusion that they both have similar functions.

- In the introduction, "while it may seem highly improbable that a few tweaks to non-coding DNA could result in a beneficial new gene". I suggest citing Vakirlis et al, Nature Communications 2020, who recently showed how few tweaks to non-coding DNA can result in proteins whose expression is beneficial for yeast.

- Line 170: "about" should be "above"

- Line 303: remove "de"

- Figure 3b: there is typo on the red paragraph. "More novel/annotated"

Oct 29 2020

Response to referees

We provide a point-by-point response to the referees. Please find our responses in red. Changes in the manuscript have also been indicated in red in the corresponding file.

Reviewer #1 (Remarks to the Author):

The work is of excellent quality and will serve as a very valuable resource. I still believe that the novel overall insight gained from this paper is limited given everything that was done on the subject before even if it presents a rich resource for annotation and biological discovery. The authors argue that the novelty from previous studies is that they focus on transcripts rather than on ORFs in the genomic sequences. However, since it has been shown that most of the yeast genome is transcribed, the two approaches are not that different.

We appreciate the comments of the reviewer. We have performed an additional analysis of the interplay between transcript evolution and ORF evolution (lines 391-410) which highlights the differences between these different levels of study.

Reviewer #3 (Remarks to the Author):

I read the resubmitted manuscript by Blevins and colleagues with great interest. I appreciate the efforts that the authors made to include more of the literature, better describe caveats, and clarify that the goal of the study was to find novel transcripts rather than de novo coding elements. These additions greatly improve the manuscript.

A. Genes vs transcripts

I am not entirely satisfied with the rephrasing to “de novo transcripts”. It is not pervasive throughout the manuscript, leading to considerable confusion (eg lines 125-130, 181, 197-204, 415...Figs2-3).

We have carefully reviewed the complete text, including the lines mentioned by the reviewer, to make sure that we use “*de novo* transcripts” instead of “*de novo* genes” for the results of our pipeline.

Furthermore, some analyses and interpretations are used that would be appropriate for ORFs/genes but not for transcripts. When the authors attempt to compare their method with others, it is no longer relevant since the goals are so different (transcript vs ORF).

We have made an effort to be as clear as possible about these different concepts. In this study we perform ribosome profiling and thus have data for the translation of the ORFs, Figure 4 is dedicated to the set of translated ORFs. The comparison with the other methods was requested by several reviewers. In order to make the results as comparable as possible we have focused on *S. cerevisiae*-specific annotated protein coding genes, which are the less affected by the methodology.

It is not a question of being more or less “conservative”. The authors should consider more deeply the interplay between ORF and transcript evolution when they want to make inferences about gene emergence. When describing the MDF1 example, they

classify the “gene” as genus specific since a homologous transcript at the same locus exists in other species. However, as Li et al. show, frameshifts and stop codons mean that no comparable intact ORF exists in these species despite the nucleotide homology.

The coding product itself is therefore species specific regardless of how old the transcript is. I do not know if the authors intend to be claiming otherwise—the paragraph is ambiguous on this point—but it should be made clear that this is not a contradiction of Li et al. More generally, in their analyses and discussions of *de novo* coding transcripts (statistics and examples), consideration should be given to ORF evolution as well. Otherwise, it is OK to write a paper about species-specific transcripts without invoking the *de novo* emergence of coding elements too.

We fully agree that the protein product of MDF1 is species specific and have made it clear by rewriting the corresponding section (lines 341-344). More generally we have investigated the interplay between the evolution of the transcript and the evolution of the ORF in a new section (lines 391-410). We have focused on *de novo* transcripts with translated ORFs in *S. cerevisiae*. We have found that in most cases the ORF is not conserved in *S. paradoxus* or *S. mikatae* even if the transcript is conserved. We discuss the results in the context of the “transcript-first” and “ORF-first” hypotheses of *de novo* gene birth.

B. Overlapping vs divergent

Regarding the finding that the species-specific transcripts tend to overlap annotated genes:

- What is the expectation? As the authors write, 70% of the genome is occupied by annotated genes. Given that the authors found novel transcripts, where would they be expected to be? Please provide simulations or statistical tests to support or refute the notion that novel transcripts are preferentially overlapping versus divergent.

Given that 70% of the genome is occupied by annotated genes we expect that a large proportion of the novel transcripts will be located antisense to other genes, which is our initial hypothesis. Studies based on SUTs/CUTs, however, reported that unannotated transcripts predominantly arose from bidirectional promoters (Xu et al., 2009; Neil et al., 2009), and similar conclusions were drawn in a study of *de novo* annotated genes (Vakirlis et al., 2018). Our study, combining phylogenetic conservation and transcriptomics data, finds that antisense *de novo* transcripts are the most abundant class. The proportion of *de novo* transcripts that are antisense to other genes is much higher than for conserved transcripts (50% vs 7.5%, Fisher test, Figure 5b). In contrast we find similar percentages for transcriptional from bidirectional promoters (27% vs 23%, respectively).

- Please be clearer in your presentation of context. A number of papers have studied unannotated/non-coding/pervasive transcripts in *cerevisiae*, although without reference to *de novo* gene birth (e.g., CUTs, SUTs, XUTs). The idea that *de novo* gene emergence may occur frequently on antisense transcripts was also shown in the past, although not explicitly for species-specific transcripts. Both of these facts should be clear from the introduction.

We are thankful for the reviewer pointing us to these studies, which are now included in the manuscript. Please refer to lines 46-48 and 428-431.

- I hope the RNA sequencing protocol used was strand -specific, otherwise this finding is completely invalidated. The methods clarify it is so for ribo-seq, but nothing is written for RNA seq. I suppose this is only an oversight in the writing of the method section.

Yes, the sequencing protocol was strand-specific. We have added this information to the methods section.

C. Additional comments

- I did not check in details, but please make sure that the phenotypes/ mechanisms described for the example ORFs in Figure 6 were not obtained by genetics. If they were, you could not distinguish whether the phenotypes are due to one or the other overlapping ORF, leading to an illusion that they both have similar functions.

They were not based on genetics, but on transcript expression inactivation/over-expression experiments, so they could distinguish between the two overlapping ORFs.

- In the introduction, “while it may seem highly improbable that a few tweaks to non-coding DNA could result in a beneficial new gene”. I suggest citing Vakirlis et al, Nature Communications 2020, who recently showed how few tweaks to non-coding DNA can result in proteins whose expression is beneficial for yeast.

Following the recommendation of the reviewer we now cite this work next to the sentence.

- Line 170: “about” should be “above”
- Line 303: remove “de”
- Figure 3b: there is typo on the red paragraph. “More novel/annotated”

We have ammended these typos.

Reviewers' Comments:

Reviewer #3:

Remarks to the Author:

The manuscript is much improved, and the added consideration of ORF-first Transcript-first brings an exciting novel angle.

Comparison to other datasets:

I appreciate the author's efforts, but the way that the figure 3 is presented just doesn't work now that the paper is about de novo transcripts rather than de novo proteins. Previous approaches were not trying to identify de novo transcripts, and this is an important distinction in goal that is still not clearly explained and dissected by the authors despite their efforts. The corresponding discussion section, lines 497-503, is misleading for the same reason. This part of the manuscripts could be removed, but the results could also be presented in a different light. Fig 3 b can be presented as further validation of the quality of the method developed by the authors, similar to the bottom panels of Fig 2d. Fig 3e can be presented to show that this dataset has good overlap with previous studies that looked at de novo gene emergence, but identifies many novel ones.

Methodological clarifications:

- For the results presented in Figs 1 d,e,f, it is unclear how the results of the 2 growth conditions are integrated. Based on the rest of the manuscript, I suppose the counts are based on translation in normal OR stress conditions, using a "union" operation, but I don't know if the translation event has to be observed in the same condition as the transcription event. This should be clarified, including and especially in Fig 1f (when there are several translated ORFs, are they translated in both conditions?)
- The text, figure 2 a,b,c,d, and method section, need clarify how transcripts are classified when they correspond to a region lacking synteny (eg, if a transcript is in a region lacking synteny in paradoxus or anywhere, does it count as de novo?)

Most de novo transcripts are overlapping in yeast

This is the main result of the paper. It is very interesting, and given the paper is framed as an investigation of where do de novo transcripts emerge in a compact genome, I would have liked to see the question more clearly answered at the end of the discussion. In particular, I feel the authors have enough data to propose that in compact genomes de novo emergence still occurs but opposite conserved genes, since there is not much other space. This leads to interesting future considerations of function, coexpression, coevolution etc of de novo emerging loci in organisms with compact versus sparse genomes. Why not end the discussion this way? It should be noted in the discussion that this result, while different from Vakirlis 2017, is in line with Carvunis 2012.

Reviewer #4:

None

Reviewer #5:

None

RESPONSE TO REVIEWERS' COMMENTS

Reviewer #3 (Remarks to the Author):

The manuscript is much improved, and the added consideration of ORF-first Transcript-first brings an exciting novel angle.

Comparison to other datasets:

I appreciate the author's efforts, but the way that the figure 3 is presented just doesn't work now that the paper is about de novo transcripts rather than de novo proteins. Previous approaches were not trying to identify de novo transcripts, and this is an important distinction in goal that is still not clearly explained and dissected by the authors despite their efforts. The corresponding discussion section, lines 497-503, is misleading for the same reason. This part of the manuscripts could be removed, but the results could also be presented in a different light. Fig 3 b can be presented as further validation of the quality of the method developed by the authors, similar to the bottom panels of Fig 2d. Fig 3e can be presented to show that this dataset has good overlap with previous studies that looked at de novo gene emergence, but identifies many novel ones.

We are now presenting the results in Fig 3 in a different light following the suggestions of the reviewer. We have modified the related text in Results (section Comparison to other approaches), as well as in the Discussion, to better reflect the differences between this approach and the previous approaches and emphasizing that, despite the differences, there is good overlap, which further validates our results.

Methodological clarifications:

- For the results presented in Figs 1 d,e,f, it is unclear how the results of the 2 growth conditions are integrated. Based on the rest of the manuscript, I suppose the counts are based on translation in normal OR stress conditions, using a "union" operation, but I don't know if the translation event has to be observed in the same condition as the transcription event.

As the reviewer hypothesized, in Figure 1d, 1e, and 1f, the counts are based on a union operation; we consider an ORF as translated if we observed a RibORF score >0.7 in either normal, stress, or both conditions, independent of the condition(s) in which transcription ≥ 15 TPM occurred. We have further clarified this in Methods (sections Prediction of translated ORFs and ORF properties).

This should be clarified, including and especially in Fig 1f (when there are several translated ORFs, are they translated in both conditions?)

In Figure 1f, each ORF is only counted once regardless of how many conditions it was translated in. For example, in the case of the 163 novel transcripts for which we detected only one translated ORF, the same ORF may have been translated above our threshold in both conditions, or only in one condition. We have clarified this in Methods (section Prediction of translated ORFs).

- The text, figure 2 a,b,c,d, and method section, need clarify how transcripts are classified when they correspond to a region lacking synteny (eg, if a transcript is in a region lacking synteny in paradoxus or anywhere, does it count as de novo?)

If the transcript corresponded to a region lacking synteny, but met the other criteria to be classified as *de novo*, it was counted as *de novo*. This affected a negligible number of transcripts because blocks of conserved synteny between pairs of species already covered 80-91% of the genome and we used multiple pairwise comparisons to define *de novo* transcripts. We have added the information on the percentage of the genome within synteny blocks in Results lines 95-96.

Most de novo transcripts are overlapping in yeast

This is the main result of the paper. It is very interesting, and given the paper is framed as an investigation of where do de novo transcripts emerge in a compact genome, I would have liked to see the question more clearly answered at the end of the discussion. In particular, I feel the authors have enough data to propose that in compact genomes de novo emergence still occurs but opposite conserved genes, since there is not much other space. This leads to interesting future considerations of function, coexpression, coevolution etc of de novo emerging loci in organisms with compact versus sparse genomes. Why not end the discussion this way? It should be noted in the discussion that this result, while different from Vakirlis 2017, is in line with Carvunis 2012.

We have modified the last paragraph of the discussion to emphasize which is the main finding of the work.